# Distributionally Robust Post-hoc Classifiers under Prior Shifts

**Jiaheng Wei**[*†]
UC Santa Cruz

**Harikrishna Narasimhan**
Google Research

**Ehsan Amid**
Google Research

**Wen-Sheng Chu**
Google Research

**Yang Liu**
UC Santa Cruz

**Abhishek Kumar** [†]
Google Research

## Abstract

The generalization ability of machine learning models degrades significantly when the test distribution shifts away from the training distribution. We investigate the problem of training models that are robust to shifts caused by changes in the distribution of class-priors or group-priors. The presence of skewed training priors can often lead to the models overfitting to spurious features. Unlike existing methods, which optimize for either the worst or the average performance over classes or groups, our work is motivated by the need for finer control over the robustness properties of the model. We present an extremely lightweight post-hoc approach that performs scaling adjustments to predictions from a pre-trained model, with the goal of minimizing a distributionally robust loss around a chosen target distribution. These adjustments are computed by solving a constrained optimization problem on a validation set and applied to the model during test time. Our constrained optimization objective is inspired from a natural notion of robustness to controlled distribution shifts. Our method comes with provable guarantees and empirically makes a strong case for distributional robust post-hoc classifiers. An empirical implementation is available at https://github.com/weijiaheng/Drops.

## 1 Introduction

Distribution shift, a problem characterized by the shift of test distribution away from the training distribution, deteriorates the generalizability of machine learning models and is a major challenge for successfully deploying these models in the wild. We are specifically interested in distribution shifts resulting from changes in marginal class priors or group priors from training to test. This is often caused by a skewed distribution of classes or groups in the training data, and vanilla empirical risk minimization (ERM) can lead to models overfitting to spurious features. These spurious features seem to be predictive on the training data but do not generalize well to the test set. For example, the background can act as a spurious feature for predicting the object of interest in images, *e.g.*, camels in a desert background, water-birds in water background (Sagawa et al., 2020).

Distributionally robust optimization (DRO) (Ben-Tal et al., 2013; Duchi et al., 2016; Duchi & Namkoong, 2018; Sagawa et al., 2020) is a popular framework to address this problem which formulates a robust optimization problem over class- or group-specific losses. The common metrics of interest in the DRO methods are either the average accuracy or the worst accuracy over classes/groups (Menon et al., 2021; Jitkrittum et al., 2022; Rosenfeld et al., 2022; Piratla et al., 2022; Sagawa et al., 2020; Zhai et al., 2021; Xu et al., 2020; Kirichenko et al., 2022). However, these metrics only cover the two ends of the full spectrum of distribution shifts in the priors. We are instead motivated by the need to measure the robustness of the model at various points on the spectrum of distribution shifts.

To this end, we consider applications where we are provided a target prior distribution (that could either come from a practitioner or default to the uniform distribution), and would like to train a

---

[*]Work done during an internship at Google Research, Brain Team.

[†]Correspondence to: Jiaheng Wei <jiahengwei@ucsc.edu>, Abhishek Kumar <abhishk@google.com>.

model that is robust to varying distribution shifts around this prior. Instead of taking the conventional approach of optimizing for either the average accuracy or the worst-case accuracy, we seek to maximize the minimum accuracy within a $\delta$-radius ball around the specified target distribution. This strategy allows us to encourage generalization on a spectrum of controlled distribution shifts governed by the parameter $\delta$. When $\delta = 0$, our objective is simply the average accuracy for the specified target priors, and when $\delta \to \infty$, it reduces to the model's worst-case accuracy, thus providing a natural way to interpolate between the two extreme goals of average and worst-case optimization.

To train a classifier that performs well on the prescribed distributionally robust objective, we propose a fast and extremely lightweight post-hoc method that learns scaling adjustments to predictions from a pre-trained model. These adjustments are computed by solving a constrained optimization problem on a validation set, and then applied to the model during evaluation time. A key advantage of our method is that it is able to reuse the same pretrained model for different robustness requirements by simply scaling the model predictions. This is contrast to several existing DRO methods that train all model parameters using the robust optimization loss (Sagawa et al., 2020; Piratla et al., 2022), which requires group annotations for the training data and requires careful regularization to make it work with overparameterized models (Sagawa et al., 2020). On the other hand, our approach only needs group annotations for a smaller held-out set and works by only scaling the model predictions of a pre-trained model at test time. Our method also comes with provable convergence guarantees. We apply our method on standard benchmarks for class imbalance and group DRO, and show that it compares favorably to the existing methods when evaluated on a range of distribution shifts away from the target prior distribution.

## 2 BACKGROUND

We are primarily interested in two specific prior shifts for distributional robustness of classifiers. In this section, we briefly introduce the problem setting of the two prior shifts and set the notation.

**Class-Level Prior Shifts.** We are interested in a multi-class classification problem with instance space $\mathcal{X}$ and output space $[m] = \{1, \ldots, m\}$. Let $\mathcal{D}$ denote the underlying data distribution over $\mathcal{X} \times [m]$, the random variables of instance $X$ and label $Y$ satisfy that $(X, Y) \sim \mathcal{D}$. We define the conditional-class probability as $\eta_y(x) = \mathbb{P}(Y = y | X = x)$ and the class priors $\pi_y = \mathbb{P}(Y = y)$, note that $\pi_y = \mathbb{E}\left[\eta_y(x)\right]$. We use $u = [\frac{1}{m}, \ldots, \frac{1}{m}]$ to denote the uniform prior over $m$ classes.

Our goal is then to learn a multi-class classifier $h : \mathcal{X} \to [m]$ that maps an instance $x \in \mathcal{X}$ to one of $m$ classes. We will do so by first learning a scoring function $f : \mathcal{X} \to \Delta_m$ that estimates the conditional-class probability for a given instance, and construct the classifier by predicting the class with the highest score: $h(x) = \arg\max_{j \in [m]} f_j(x)$. We measure the performance of a scoring function using a loss function $\ell : [m] \times \Delta_m \to \mathbb{R}_+$ and measure the per-class loss using $\ell_i(f) := \mathbb{E}\left[\ell(y, f(x)) \,|\, y = i\right]$.

Let $\{(x_i, y_i)\}_{i=1}^n$ be a set of training data samples. The empirical estimate of training set prior is $\hat{\pi}_i := \frac{1}{n} \sum_{j \in [n]} \mathbf{1}(y_j = i)$ where $\mathbf{1}(\cdot)$ is the indicator function. In class prior shift, the class prior probabilities at test time shift away from $\hat{\pi}$. A special case of such class-level prior shifts includes class-imbalanced learning (Lin et al., 2017; Cui et al., 2019; Cao et al., 2019; Ren et al., 2020; Menon et al., 2021) where $\hat{\pi}$ is a long-tailed distribution while the class prior at test time is usually chosen to be the uniform distribution. Regular ERM tends to focus more on the majority classes at the expense of ignoring the loss of the minority classes. Recent work (Menon et al., 2021) uses temperature-scaled logit adjustment with training class priors to adapt the model for average class accuracy. Our method also applies post-hoc adjustments to model probabilities, but our goal differs from (Menon et al., 2021) as we care for varying distribution shifts around the uniform prior and the scaling adjustments are learned using a held-out set to optimize for a constrained robust loss.

**Group-Level Prior Shifts.** The notion of *groups* arises when each data point $(x, y)$ is associated with some attribute $a \in A$ that is spuriously correlated with the label. This is used to form $m = |A| \times |Y|$ groups as the cross-product of $|A|$ attributes and $|Y|$ classes. The data distribution $\mathcal{D}$ is taken to be a mixture of $m$ groups with mixture prior probabilities $\pi$, and each group-conditional distribution given by $\mathcal{D}_j, j \in [m]$. In this scenario, we have $n$ training samples $\{(x_i, y_i)\}_{i=1}^n$ drawn i.i.d. from $\mathcal{D}$, with empirical group prior probabilities $\hat{\pi}$. For skewed group-prior probabilities $\hat{\pi}$, regular ERM is vulnerable to spurious correlations between the attributes and labels, and the accuracy degrades

when the test data comes from a shifted group prior (*e.g.*, balanced groups). Domain-aware methods typically assume that the attributes are available for the training examples and optimize for the worst or average group loss (Sagawa et al., 2020; Piratla et al., 2022). However, recent work (Rosenfeld et al., 2022; Kirichenko et al., 2022) has observed that ERM on skewed group priors is able to learn core features (in addition to spurious features), and training a linear classifier on top of ERM learned features with a small balanced held-out set works quite well. Our proposed method is aligned with this recent line of work in assuming access to only a small held-out set with group annotations. However, we differ in two aspects: **(i)** Our method is more lightweight and works by only scaling the model predictions post-hoc during test time, **(ii)** The scaling adjustments are learned to allow more control over the desired robustness properties than implicitly targeting the worst or average accuracies as in (Kirichenko et al., 2022).

**Evaluation Metrics under Prior Shifts.**   Typical evaluation metrics used under prior shifts are:
- Mean: This evaluation metric assigns uniform weights for each class or group, measuring the average class- or group-level test accuracy.
- Worst: This evaluation metric measures the performance on the worst class or group, and fails to examine the effectiveness of proposed methods on the remaining classes.
- Stratified: To overcome the issue of worst-case metric, stratified metrics are sometimes used that divide the classes into three strata and report average accuracy in each stratum, specifically (i) Head: average accuracy on subsets of classes that contain more than a specified number (*e.g.*, 100) of training samples; (ii) Torso: average accuracy on classes that contain, *e.g.*, 20 to 100 samples; (iii) Tail: average accuracy on tail classes.

The *mean* and *worst* metrics are limited in that they probe the model robustness only on the two ends of the full spectrum. The *stratified* metric looks into three strata of classes, but the design of strata is more heuristic and doesn't allow for a principled interpolation between *mean* and *worst* metrics.

## 3   DISTRIBUTIONAL ROBUSTNESS OBJECTIVE UNDER PRIOR SHIFTS

We assume that we are given a target prior distribution (that could either come from a practitioner or default to the uniform distribution), and our goal is to train a model that is robust to varying distribution shifts around this prior. To this end, we consider an objective that naturally captures this goal and is based on the weighted sum of class-level (or group-level) performances on the test data, *i.e.*, $\sum_{i \in [m]} g_i \text{Acc}_i$, where $\text{Acc}_i$ indicates the accuracy of the class/group $i$:

$$\delta\text{-worst accuracy:} \qquad \min_{g \in \Delta_m} \sum_{i \in [m]} g_i \text{Acc}_i, \quad \text{s.t. } D(g, r) \leq \delta \,. \tag{1}$$

Here $\Delta_m$ denotes the $(m-1)$-dimensional probability simplex and $r \in \Delta_m$ is a specified reference (or target) distribution. The $\delta$-worst accuracy seeks the worst-case $g$-weighted performance with the weights constrained to lie within the $\delta$-radius ball (defined by the divergence $D : \Delta_m \times \Delta_m \to \mathbb{R}$) around the target distribution $r$. For uniform distribution $r = u$ and any choice of divergence $D$, it reduces to the mean accuracy for $\delta = 0$ and the worst accuracy for $\delta \to \infty$. The objective interpolates between these two extremes for other values of $\delta$ and captures our goal of optimizing for variations around target priors instead of more conventional objectives of optimizing for either the average accuracy at the target prior or the worst-case accuracy. The divergence constraint in the $\delta$-worst objective is convex in $g$ for several divergence functions (*e.g.*, $f$-divergence, Bregman divergence), and allows for efficient measurement as well (if the per-class accuracies are known) using standard packages such as CVXPY (Diamond & Boyd, 2016).

## 4   ROBUST POST-HOC CLASSIFIERS UNDER CLASS PRIOR SHIFTS

In this section, we propose a **D**istributionally **RO**bust **P**o**S**t-hoc method, **DROPS**, which enables the reuse of a pre-trained model for different robustness requirements by simply scaling the model predictions. In alignment with the earlier $\delta$-worst objective, we aim to learn a classifier $h$ with scoring function $f$ as follows:

$$\textbf{Goal:} \qquad \min_{f : \mathcal{X} \to \Delta_m} \max_{g \in \Delta_m} \sum_{i \in [m]} g_i \, \mathbb{P}_{X|Y=i}(f(X) \neq Y), \ \text{s.t. } D(g, r) \leq \delta \,. \tag{2}$$

### 4.1 BAYES-OPTIMAL SCORER

Recall that we measure the performance of a scoring function using a loss function $\ell : [m] \times \Delta_m \to \mathbb{R}_+$, and the per-class loss using $\ell_i(f) := \mathbb{E}\left[\ell(y, f(x)) \,|\, y = i\right]$. To facilitate theoretical analysis, we consider the scenario where the target distribution $r$ is the uniform prior $u = [\frac{1}{m}, \ldots, \frac{1}{m}]$. We are then interested in minimizing the following robust objective optimization:

$$\min_{f:\mathcal{X}\to\Delta_m} \mathrm{DRL}(f;\delta) = \min_{f:\mathcal{X}\to\Delta_m} \max_{g\in\mathbb{G}(\delta)} \sum_{i=1}^{m} g_i \, \ell_i(f)\,, \quad (3)$$

where $\mathbb{G}(\delta) = \{g \in \Delta_m \,|\, D(g, u) \le \delta\}$ for some $\delta > 0$ and divergence function $D : \Delta_m \times \Delta_m \to \mathbb{R}_+$. We first derive the Bayes-optimal solution to the learning problem in equation 3.

**Theorem 1** (Bayes-Optimal scorer). *Suppose $\ell(y, z)$ is a proper loss that is convex in its second argument and $D(g, \cdot)$ is convex in $g$. Let $\delta > 0$ be such that $\mathbb{G}(\delta)$ is non-empty. The optimal solution to equation 3 takes the following form for some $g^* \in \mathbb{G}(\delta)$:*

$$f_y^*(x) \propto \frac{g_y^*}{\pi_y} \cdot \eta_y(x)\,.$$

The proof is provided in Appendix A.1 and would use in its intermediate step, the following standard result for cost-sensitive learning:

**Lemma 2.** *Suppose $\ell(y, z)$ is a proper loss and $\ell_i(f) = \mathbb{E}\left[\ell(y, f(x)) \,|\, y = i\right]$. For any fixed $g \in \Delta_m$, the following is the minimizer to the objective $\sum_{i=1}^{m} g_i \cdot \ell_i(f)$ over all measurable functions $f : \mathcal{X} \to \Delta_m$: $f_y^*(x) \propto \frac{g_y}{\pi_y} \cdot \eta_y(x)$.*

### 4.2 POST-HOC APPROACH

Given a pre-trained model $\hat{\eta} : \mathcal{X} \to \Delta_m$ that estimates the conditional-class probability function $\eta$, we seek to approximate the Bayes-optimal classifier. To do so, we write the Lagrangian form of equation 3 as: $\mathcal{L}(f, g, \lambda) = \sum_{i=1}^{m} g_i \cdot \ell_i(f) - \lambda(D(g, u) - \delta)$. We next show the optimization w.r.t. the Lagrangian form has the following equivalent unconstrained problem, as specified in Proposition 1.

$$\min_{f:\mathcal{X}\to\Delta_m} \max_{g\in\Delta_m} \min_{\lambda\ge0} \mathcal{L}(f, g, \lambda)\,. \quad (4)$$

**Proposition 1.** *The equivalence of two optimization tasks:*

$$\min_{f:\mathcal{X}\to\Delta_m} \mathrm{DRL}(f;\delta) \quad \Longleftrightarrow \quad \min_{f:\mathcal{X}\to\Delta_m} \max_{g\in\Delta_m} \min_{\lambda\ge0} \mathcal{L}(f, g, \lambda)\,.$$

To understand why the above two optimization tasks are equivalent, we point out that if the constraint is violated, then the minimizer over $\lambda$ would yield an unbounded objective. The maximizer over $g$, as a result, will always choose a $g$ that satisfies the constraint.

Build upon Proposition 1, one can then solve the equivalent min-max problem by alternating between a gradient-descent update on $\lambda$, an Exponentiated Gradient (EG)-ascent update (Kivinen & Warmuth, 1997) on $g$, and a full minimization over $f$: $\min_{f:\mathcal{X}\to\Delta_m} \sum_{i=1}^{m} g_i \cdot \ell_i(f)$, the optimal solution for which is given from Lemma 2 by $f_y^*(x) \propto \frac{g_y}{\pi_y} \cdot \eta_y(x)$. Thus, for the pre-trained model $\hat{\eta}$, we propose to make post-hoc adjustments over $\hat{\eta}$ to make the classifiers more robust to prior shifts. We construct the "post-shifted" classifier by predicting the class with the "post-shifted" highest score for each $x$:

$$\textbf{DROPS:} \quad h(x) \in \arg\max_{i\in[m]} \frac{g_i}{\hat{\pi}_i} \cdot \hat{\eta}_i(x)\,.$$

Intuitively, model prediction of class $i$ is up-scaled if the class $i$ is assigned with a large weight $g_i$, or is of a small prior $\hat{\pi}_i$.

**An Empirical Alternative.** Given a sample $x$, the "post-shifted" classifier takes the form $h(x) \in \arg\max_{i\in[m]} \frac{g_i}{\hat{\pi}_i} \cdot \hat{\eta}_i(x)$. The softmax activation function then maps the per-class mode prediction

logit, denoted by $\text{Logit}_i(x)$, into the corresponding probability. Specifically, for $i \in [m]$, we have $\hat{\eta}_i(x) = \frac{\exp(\text{Logit}_i(x))}{\sum_{j \in [m]} \exp(\text{Logit}_j(x))}$. Thus, an empirical alternative of the proposed DROPS is a logit-adjustment approach where, given a sample $x$, for fixed class weights $\frac{g_i}{\hat{\pi}_i}$, we have:

$$\textbf{DROPS: } h(x) \in \text{argmax}_{y \in [m]} \frac{g_y}{\hat{\pi}_y} \cdot \hat{\eta}_y(x) \iff h(x) \in \text{argmax}_{y \in [m]} \text{Logit}_y(x) + \log(\frac{g_y}{\hat{\pi}_y}).$$

### 4.3 Convergence of Post-hoc Classifier

In practice, we only have access to an empirical estimate of the Lagrangian form computed using a validation set $S = \{(\tilde{x}_1, \tilde{y}_1), \ldots, (\tilde{x}_n, \tilde{y}_n)\}$:

$$\hat{\mathcal{L}}(f, g, \lambda) = \sum_{i=1}^{m} g_i \cdot \hat{\ell}_i(f) - \lambda(D(g, u) - \delta), \tag{5}$$

where $\hat{\ell}_i = \frac{1}{\hat{\pi}_i} \sum_{(x,y) \in S: y=i} \ell(y, f(x))$ and $\hat{\pi}_i = \frac{1}{n}|\{(x,y) \in S : y = i\}|$. We can then approximately solve the saddle-point problem in equation 4 by repeating the following three steps for $T$ number of iterations:

**Step 1: updating $\lambda^{(t)}$.** Given the step size $\eta_\lambda > 0$, we perform gradient updates on $\lambda$ through:

$$\lambda^{(t+1)} = \lambda^{(t)} - \eta_\lambda \nabla \mathcal{L}(f^{(t)}, g^{(t)}, \lambda^{(t)}) = \lambda^{(t)} - \eta_\lambda(\delta - D(g^{(t)}, u)).$$

For KL divergence, the updated $\lambda^{(t+1)}$ amounts to $\lambda^{(t+1)} \leftarrow \lambda^{(t)} - \eta_\lambda(\delta - \sum_i g_i^{(t)} \log(\frac{g_i^{(t)}}{u_i}))$. We can clip $\lambda$ if the update falls out of the desired range: $\lambda^{(t+1)} \leftarrow \text{clip}_{[0,R]} \left( \lambda^{(t)} - \eta_\lambda(\delta - D(g, u)) \right)$ for some $R > 0$.

**Step 2: updating $g^{(t)}$.** Given the step size $\eta_g > 0$, we perform gradient updates on $g$ through:

$$g_i^{(t+1)} \leftarrow g_i^{(t)} \cdot \exp\left( \eta_g \cdot \nabla_{g_i} \hat{\mathcal{L}}(f^{(t)}, g^{(t)}, \lambda^{(t)}) \right).$$

To obtain the updates for $g$, we consider the fixed $\lambda^{(t)} \in \mathbb{R}_+$ and adopt a simplified version as $g_i^{(t+1)} \leftarrow u_i \cdot \exp\left( \hat{\ell}_i(f^{(t)})/\lambda^{(t)} \right)$ empirically.

**Step 3: scaling the predictions.** In this step, we apply the post-hoc shifts on the classifier's prediction through $f_y^{(t+1)}(x) \propto \frac{g_y^{(t+1)}}{\hat{\pi}_y} \cdot \hat{\eta}_y(x)$, in which we use the pre-trained estimate $\hat{\eta}$ of the conditional-class probabilities.

The final scorer returned is an average of scorers from each iteration: $\bar{f}(x) = \frac{1}{T} \sum_{t=1}^{T} f^{(t)}(x)$.

We now provide a convergence guarantee for this procedure. For a fixed pre-trained model $\hat{\eta} : \mathcal{X} \to \Delta_m$, we denote the class of post-hoc adjusted functions derived from it by

$$\mathcal{F} = \left\{ f : x \mapsto \left( \frac{\beta_1 \hat{\eta}_1(x)}{\sum_j \beta_j \hat{\eta}_j(x)}, \ldots, \frac{\beta_m \hat{\eta}_m(x)}{\sum_j \beta_j \hat{\eta}_j(x)} \right) \mid \beta \in \Delta_m \right\},$$

and we let $|\mathcal{F}|$ denote a measure of complexity of this function class.

**Theorem 3.** *Fix $\delta > 0$. Suppose $\ell(y, z)$ be a proper loss that is convex and $L_\ell$-Lipschitz in its second argument, and $\ell(y, z) \leq B_\ell$, for some $B_\ell > 0$. Suppose $D(g, u) \leq B_D$, for some $B_D > 0$, and $D$ is convex and $L_D$-Lipschitz in its first argument. Furthermore, suppose $\max_{y \in [m]} \frac{1}{\pi_y} \leq Z$, for some $Z > 0$. Then when $n \geq 8Z \log(2m/\alpha)$, and we set $T = \mathcal{O}(n)$, $R = 2B_\ell Z/\delta$, $\eta_\lambda = \frac{R}{B_D \sqrt{n}}$ and $\eta_g = \frac{1}{2B_\ell Z + RL_D} \sqrt{\frac{\log(m)}{n}}$, we have with probability at least $1 - \alpha$ over draw of validation sample $S \sim \mathcal{D}^n$, the classifier returned $\bar{f}(x) = \frac{1}{T} \sum_{t=1}^{T} f^{(t)}(x)$ satisfies:*

$$\text{DRL}(\bar{f}; \delta) \leq \min_{f : \mathcal{X} \to \Delta_m} \text{DRL}(f; \delta) + \mathcal{O}\left( \sqrt{\frac{\log(m|\mathcal{F}|/\alpha)}{n}} + \mathbb{E}_x \left[ \|\hat{\eta}(x) - \eta(x)\|_1 \right] \right).$$

The proof is provided in Appendix A.3. The complexity measure $|\mathcal{F}|$ can be further bounded using, for example, standard covering number arguments (Shalev-Shwartz & Ben-David, 2014). Notice that the convergence to the optimal classifier is bounded by two terms: the first is a sample complexity term that goes to 0 as the number of validation samples $n \to \infty$; the second term measures how well the pre-trained model $\hat{\eta}$ is calibrated, i.e., how well its scores match the underlying conditional-class probabilities.

### 4.4 ROBUST POST-HOC CLASSIFIERS UNDER GROUP PRIOR SHIFTS

We now introduce the variant of our approach to address group prior shifts. We are interested in the performance of the trained classifiers with additional attribute information available in a held-out validation set. Note that the settings of class and group prior shifts differ in the knowledge of group information at validation and test time. Thus, the first variant of DROPS under group prior shifts is the same as the class prior setting: DROPS completely ignores the per-example group information and post-shifts the model predictions using only class labels. We consider another variant where we have access to the attribute information during both validation and test. This scenario can actually arise in practice when the attribute information is readily available for test examples, such as device or sensor type in the case of data coming from different devices/sensors. In this case, the attribute-specific class priors take the form of $\pi_{a,i} = \mathbb{P}(y = i|a)$. DROPS can be naturally adapted to this setting by learning multiple sets of scaling adjustments, one for each attribute type, and using the scaling adjustment corresponding to the attribute of the test example at prediction time. We provide the discussion, and the form of the Bayes-optimal classifier for this setting in Appendix B.

## 5 RELATED WORK

**Class-Imbalanced Learning.** Most work in class-imbalanced learning is typically interested in generalizing on a uniform class prior when the training data has a skewed or imbalanced class prior. Existing solutions to the class imbalanced learning problem could be categorized into three major lines: (1) Information augmentation methods, which make use of additional information such as open set data (Wei et al., 2021), adopt a transfer learning approach to enrich the representation on the tail classes (Liu et al., 2020; Yin et al., 2019), or use advanced data-augmentation techniques (Perez & Wang, 2017; Shorten & Khoshgoftaar, 2019); (2) Module improvement methods, *i.e.*, the decoupled training on the head/tail classes (Kang et al., 2019; Chu et al., 2020; Zhong et al., 2021), or through an ensemble way to make use of multiple networks with different expertise/concentration (Zhou et al., 2020; Guo & Wang, 2021; Wang et al., 2020); (3) The most related to our work are the class re-balancing based methods, which mitigate the impact of class-imbalanced data by adjusting the logits using the class prior (Menon et al., 2021), or align the distributions of the model prediction and a set of balanced validation set (Zhang et al., 2021b), or modify the loss values by referring to the label frequency (Ren et al., 2020), sample influence (Park et al., 2021), among many other robust loss designs (Amid et al., 2019; Wei & Liu, 2021; Zhu et al., 2021) and re-weighting schemes (Kumar & Amid, 2021; Cheng et al., 2021; Wei et al., 2022). Label shift is a related problem setting where the training class prior is not so imbalanced but the class prior shifts during test time with $p(x|y)$ staying the same, and the goal is to mitigate the effect of this shift (Lipton et al., 2018; Azizzadenesheli et al., 2019; Alexandari et al., 2020). However, our goal differs from these methods as we are primarily interested in generalization at worst case variations around the target distribution.

**Group Distributional Robustness.** It has been observed that classifiers learned with regular ERM are vulnerable to spurious correlations between the attributes and labels, and tend to perform worse when the test data comes from a shifted group prior. Most methods for group distributional robustness are interested in the average or worst group performance. Several prior works utilize the group information at the training time (Sagawa et al., 2020; Piratla et al., 2022). Recent works consider a more practical setting where the classifier does not have access to the group information at the training time, *i.e.*., data re-balancing (Idrissi et al., 2022) or re-weighting high loss examples (Liu et al., 2021), or logit-correction (Liu et al., 2023), vision transformer (ViT) models (Ghosal et al., 2022). It was also observed recently that ERM is able to learn features that can be reused for group robustness by training a linear classifier using a balanced held-out set (Kirichenko et al., 2022; Rosenfeld et al., 2022). Most relevant to us is AdvShift (Zhang et al., 2021a), which mitigates the

impact of label shifts by optimizing a distributionally robust objective function with respect to the model parameters. Out work is in similar vein in assuming access to only a held-out set with group annotations, however our method is more lightweight and allows to optimize for varying worst-case perturbations around the target distribution of interest using only post-hoc scaling of model predictions. Other related work includes CGD (Piratla et al., 2022) that proposes a learning paradigm for optimizing the average group accuracy, and Invariant risk minimization (IRM) (Arjovsky et al., 2019; Ahuja et al., 2020) that aims to learn core features using data from multiple environments for mitigating the spurious correlations.

## 6 Experiments

In this section, we empirically demonstrate the effectiveness of our proposed method DROPS, for the tasks of class-imbalanced learning and group distributional robustness.

### 6.1 Experiments on Class-imbalanced Learning

We consider the class-imbalanced task to illustrate the robustness of DROPS under class prior shifts. For CIFAR-10, and CIFAR-100 datasets, we down-sample the number of samples for each class to simulate the class-imbalance as done in earlier works (Cui et al., 2019; Cao et al., 2019). We define the imbalance ratio as $\rho := \frac{\max_{y \in [m]} \pi_y}{\min_{y \in [m]} \pi_y}$ and experiment with $\rho \in [10, 50, 100]$.

To demonstrate the effectiveness of DROPS, we compare the performance of our proposed method with several popular class-imbalanced learning approaches, including: Cross-Entropy (CE) loss, Focal Loss (Lin et al., 2017), Class-Balanced (CB) loss (Cui et al., 2019), LDAM (Cao et al., 2019), Balanced-Softmax (Ren et al., 2020), Logit-adjustment (Menon et al., 2021), and AdvShift (Zhang et al., 2021a) on the original test data. All methods are trained with the same architecture (PreAct-ResNet 18 (He et al., 2016)) with 5 random seeds, same data augmentation techniques, the same SGD optimizer with a momentum of 0.9 with Nesterov acceleration. All methods share the same initial learning rate of 0.1 and a piece-wise constant learning rate schedule of $[10^{-2}, 10^{-3}, 10^{-4}]$ at $[30, 80, 110]$ epochs, respectively. We use a batch size of 128 for all methods and train the model for 140 epochs. A balanced held-out validation set is utilized for hyper-parameter tuning. All baseline models are picked by referring to the $\delta = 1.0$-worst case performances on the validation set, which is made up of the last 10% of the original CIFAR training dataset. DROPS obtains the optimal post-hoc shifts under a variety of $\delta_{\text{train}}$ parameters, which is the perturbation hyper-parameter $\delta$ in the Lagrangian of equation 4. We take the divergence $D$ to be the KL-divergence.

**Performances Comparisons on the Generic $\delta$-Worst Case Accuracy.** Experiment results in Table 1 demonstrate that DROPS can not only give promising performance under the reported $\delta$-worst case accuracy (for $\delta = 1.0$), it also outperforms other methods in the worst case accuracy, and remains competitive on the mean accuracy as well (performs best on the CIFAR-10 setting).

To further investigate the robustness of each method under different level of prior shifts, we visualize the performance of each method under a list of $\delta$ within the $\delta$-worst case accuracy. For the uniform distribution $r = u$, the $\delta$ values recover both the mean accuracy (with $\delta = 0$) and the worst accuracy (for large enough $\delta$), and interpolates between these two metrics for other values of $\delta$. For DROPS, we train using $\delta_{\text{train}} \in [0.5, 1.0, 1.5, 2.0]$ (CIFAR-10) and $\delta_{\text{train}} \in [0.5, 1.0, 2.0, 3.0, 4.0]$ (CIFAR-100) with KL-divergence while evaluating for the aforementioned range of $\delta_{\text{eval}}$ values. We also evaluate using Reverse-KL divergence to examine the behavior under a different divergence function from that used in learning the scaling adjustments with DROPS. Figure 1 illustrates the robustness and effectiveness of our proposed DROPS method: specifically, in each sub-figure of Figure 1, the $x$-axis indicates the value of $\delta$, and the $y$-axis denotes the corresponding $\delta$-worse case accuracy. For experiment results on CIFAR-10 datasets (1st row), the curves of DROPS under different $\delta_{\text{train}}$ are consistently higher than the other baselines, indicating that with the increasing of perturbation level of the distribution shifts (from left to right in each sub-figure), DROPS is more robust to the distribution shift. As for CIFAR-100 dataset, we do observe that optimizing for the controlled worst case performance may lead to a trade-off in the worst case accuracy and averaged test accuracy, i.e., Logit-Adj is more competitive than DROPS in the measure of the averaged accuracy, while DROPS still suffers less from the increase of perturbation level.

Table 1: Performance comparisons on class-imbalanced CIFAR datasets: mean $\pm$ std of averaged class accuracy, $\delta = 1.0$-worst case accuracy, and worst class accuracy of 5 runs are reported. Best performed methods (corresponds to the averaged accuracy of 5 runs) in each setting are highlighted in purple. And we defer the paired student t-test between each baseline method and DROPS in Appendix (Table 4), to demonstrate the robustness of DROPS.

| Method | CIFAR-10 (Averaged) | | | CIFAR-100 (Averaged) | | |
|---|---|---|---|---|---|---|
| | $\rho = 10$ | $\rho = 50$ | $\rho = 100$ | $\rho = 10$ | $\rho = 50$ | $\rho = 100$ |
| Cross Entropy | 86.51±0.17 | 75.74±0.97 | 69.28±0.94 | 55.59±0.50 | 43.39±0.76 | 37.96±0.27 |
| Focal | 75.34±4.65 | 58.55±3.34 | 46.23±2.11 | 51.52±0.93 | 32.68±0.75 | 26.53±0.74 |
| LDAM | 86.63±0.33 | 76.00±0.78 | 68.81±1.01 | 55.42±0.67 | 43.03±0.68 | 38.35±0.63 |
| Balanced-Softmax | 87.18±0.24 | 78.76±0.44 | 35.55±0.35 | 56.37±0.54 | 45.10±0.51 | 41.21±0.23 |
| AdvShift | 86.33±0.21 | 74.74±0.54 | 69.41±0.83 | 55.38±0.47 | 42.71±0.96 | 38.44±0.58 |
| Logit-Adjust | 89.01±0.30 | 82.60±0.32 | 77.49±0.41 | 59.18±0.30 | 47.19±1.33 | 43.03±0.94 |
| Logit-Adj (post-hoc) | 88.98±0.29 | 82.48±0.75 | 77.94±0.50 | 59.33±0.51 | 47.77±1.49 | 43.68±1.08 |
| **DROPS** ($\delta = 0.9$) | 89.17±0.24 | 83.12±0.45 | 80.15±0.50 | 59.69±0.39 | 47.83±0.80 | 43.20±0.78 |
| Method | CIFAR-10 ($\delta = 1.0$-worst case acc) | | | CIFAR-100 ($\delta = 1.0$-worst case acc) | | |
| | $\rho = 10$ | $\rho = 50$ | $\rho = 100$ | $\rho = 10$ | $\rho = 50$ | $\rho = 100$ |
| Cross Entropy | 79.98±0.13 | 59.48±1.75 | 43.30±3.63 | 27.06±0.53 | 11.24±0.97 | 6.52±0.18 |
| Focal | 64.96±5.67 | 43.04±5.05 | 30.46±3.13 | 23.62±1.19 | 6.32±0.57 | 3.42±0.24 |
| LDAM | 79.68±0.61 | 59.58±3.06 | 42.94±2.42 | 27.44±1.56 | 11.26±0.60 | 6.62±0.71 |
| Balanced-Softmax | 80.72±0.51 | 68.34±0.55 | 24.10±0.33 | 30.36±0.81 | 17.06±0.47 | 12.00±0.52 |
| AdvShift | 80.70±0.43 | 60.36±0.90 | 48.18±1.91 | 29.37±0.94 | 14.65±0.62 | 10.71±0.82 |
| Logit-Adjust | 82.18±0.46 | 73.50±1.09 | 63.62±1.70 | 28.02±1.83 | 15.86±1.26 | 11.66±1.40 |
| Logit-Adj (post-hoc) | 82.52±0.47 | 73.84±1.69 | 64.54±1.33 | 33.46±1.26 | 17.96±1.08 | 14.34±0.53 |
| **DROPS** ($\delta = 0.9$) | 86.20±0.34 | 79.40±0.57 | 75.46±0.44 | 44.96±0.52 | 30.12±0.66 | 25.58±0.50 |
| Method | CIFAR-10 (Worst) | | | CIFAR-100 (Worst) | | |
| | $\rho = 10$ | $\rho = 50$ | $\rho = 100$ | $\rho = 10$ | $\rho = 50$ | $\rho = 100$ |
| Cross Entropy | 78.29±0.86 | 55.32±3.58 | 36.00±6.11 | 9.79±2.57 | 1.38±0.91 | 0.00±0.00 |
| Focal | 63.69±5.99 | 40.73±4.98 | 28.16±3.94 | 8.65±3.58 | 0.00±0.00 | 0.00±0.00 |
| LDAM | 76.87±0.99 | 56.57±3.18 | 35.99±2.72 | 10.57±2.65 | 1.82±1.16 | 0.27±0.50 |
| Balanced-Softmax | 78.86±0.84 | 67.11±0.98 | 22.63±0.29 | 16.17±3.85 | 6.05±2.24 | 2.58±1.42 |
| AdvShift | 79.60±0.93 | 57.71±1.60 | 44.70±2.49 | 13.38±2.36 | 3.18±1.29 | 2.05±1.32 |
| Logit-Adjust | 80.54±1.11 | 71.24±1.72 | 61.18±2.07 | 15.99±3.52 | 5.44±1.31 | 1.87±1.21 |
| Logit-Adj (post-hoc) | 81.41±1.24 | 72.00±1.72 | 63.16±2.43 | 16.21±3.93 | 4.47±1.71 | 3.39±1.46 |
| **DROPS** ($\delta = 0.9$) | 82.22±1.30 | 76.33±0.89 | 71.62±1.34 | 25.98±3.95 | 11.65±3.05 | 8.61±2.22 |

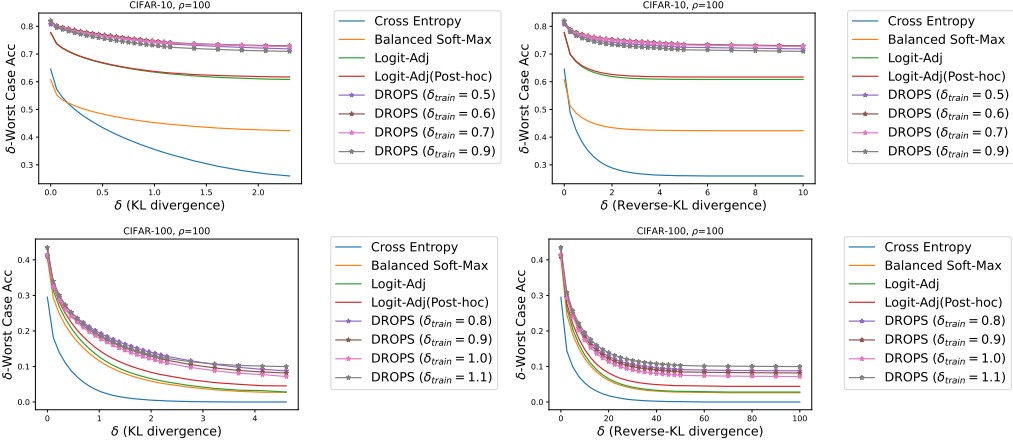

Figure 1: Performance/Robustness of methods under different perturbation $\delta$-worst case accuracy. (1st row: CIFAR-10 with imbalance ratio $\rho = 100$, adopt the KL-divergence measure (left) and Reverse-KL divergence (right) for the $\delta$-worst case accuracy calculation; 2nd row: CIFAR-100 with imbalance ratio $\rho = 100$, adopt the KL-divergence measure (left) and Reverse-KL divergence (right) for the $\delta$-worst case accuracy calculation.)

## 6.2 EXPERIMENTS ON GROUP ROBUSTNESS TASKS

We consider the group robustness tasks to show the robustness of DROPS under group prior shifts. For waterbirds and CelebA datasets, we obtain the datasets as described in (Sagawa et al., 2020).

We compare the performance of our proposed method with several popular group-Dro approaches, i.e., Just-train-twice (JTT) (Liu et al., 2021), Group Distributional Robustness (G-DRO) (Sagawa

et al., 2020), data balancing strategies (SUBG) (Idrissi et al., 2021), last-layer re-training (DFR) (Kirichenko et al., 2022). Among the baseline methods, ERM, JTT, G-DRO, and SUBG are trained with the same architecture (ResNet 50 (He et al., 2016)) for 5 runs, the base model for DFR and our approach DROPS also used the same ERM pre-trained model. Note that the knowledge requirement over group information and validation set differs among the reported methods, we clarify the differences in the column "Group Info" of Table 2.

As for the performance of DROPS in the group DRO datasets, since there are only two classes within the two datasets, we define a single post-hoc scalar as $w$, which is considered to be a hyperparameter by scaling the classifier's prediction on class 1. Note that the spurious features (minority groups among each class) tend to have a significant impact on the ERM performances by referring to the imbalanced group distribution. We use DROPS to learn post-doc corrections for both ERM trained model and for the DFR (Kirichenko et al., 2022) model that additionally retrains the last layer of the ERM-trained model with a balanced held-out set. We refer to the post-hoc corrected DFR as DROPS* in the result tables.

**Performance Comparisons on various $\delta$-Worst Accuracies.** Post-hoc scaling on the ERM model (referred as DROPS in Table 2) improves the robustness of the model as measured at various $\delta$ values (including $\delta = 0$ and $\delta \to \infty$). DROPS also outperforms JTT on the CelebA dataset. When applying post-hoc scaling to a better pre-trained model, i.e., with DFR (Kirichenko et al., 2022) that re-trains the last layer on the validation set, DROPS* outperforms all baseline methods in most settings, with the performance improvement especially clear on the CelebA dataset.

Table 2: Performance comparisons on Waterbirds and CelebA: averaged accuracy with training prior weight and uniform weight, $\delta$-worst accuracy for several $\delta$ perturbations, and worst group-level test accuracy are reported. The *Group Info* column indicates whether the group labels are available to the methods on train/validation sets. The symbol ✓✓ means the method also re-trains the last layer on the validation set (w/o needing the group information for training).

| Method | Group Info | | (Train prior) | *Waterbirds* ($r \to$ uniform prior) | | | | | |
| | Train | Val | Averaged | Averaged | $\delta = 0.05$-**Worst** | $\delta = 0.1$-**Worst** | $\delta = 0.2$-**Worst** | $\delta = 0.5$-**Worst** | Worst |
|---|---|---|---|---|---|---|---|---|---|
| ERM | ✗ | ✓ | 98.08±0.20 | 88.09±0.90 | 84.31±1.24 | 82.71±1.41 | 80.51±1.68 | 76.65±2.25 | 70.65±3.32 |
| JTT | ✗ | ✓ | 93.05±0.36 | 89.56±0.69 | 88.59±0.74 | 88.24±0.75 | 87.81±0.77 | 87.13±0.85 | 86.18±1.08 |
| **DROPS** | ✗ | ✓ | 97.95±0.16 | 88.55±1.15 | 85.50±1.51 | 84.33±1.64 | 82.81±1.79 | 80.41±2.00 | 77.14±2.15 |
| G-DRO | ✓ | ✓ | 93.03±0.34 | 91.67±0.22 | 91.23±0.33 | 91.06±0.34 | 90.83±0.40 | 90.44±0.52 | 89.85±0.73 |
| SUBG | ✓ | ✓ | 91.97±0.50 | 90.05±0.44 | 89.46±0.40 | 89.24±0.39 | 88.98±0.40 | 88.59±0.49 | 88.12±0.76 |
| $DFR_{Tr}^{Tr}$ | ✓ | ✓ | 95.83±0.94 | 93.45±0.49 | 92.77±0.48 | 92.51±0.51 | 92.16±0.55 | 91.58±0.67 | 90.72±0.91 |
| $DFR_{Tr}^{Val}$ | ✗ | ✓✓ | 93.17±1.30 | 93.29±0.80 | 92.98±0.84 | 92.86±0.85 | 92.70±0.87 | 92.42±0.88 | 92.01±0.88 |
| **DROPS*** | ✗ | ✓✓ | 93.01±1.32 | 93.42±0.61 | 93.08±0.81 | 92.98±0.90 | 92.76±1.02 | 92.45±1.23 | 91.99±1.56 |

| Method | Group Info | | (Train prior) | *CelebA* ($r \to$ uniform prior) | | | | | |
| | Train | Val | Averaged | Averaged | $\delta = 0.05$-**Worst** | $\delta = 0.1$-**Worst** | $\delta = 0.2$-**Worst** | $\delta = 0.5$-**Worst** | Worst |
|---|---|---|---|---|---|---|---|---|---|
| ERM | ✗ | ✓ | 95.33±0.12 | 81.18±1.60 | 73.78±2.36 | 70.53±2.69 | 65.97±3.15 | 57.82±3.98 | 44.89±5.30 |
| JTT | ✗ | ✓ | 87.78±0.73 | 85.04±1.05 | 83.34±1.35 | 82.89±1.49 | 82.35±1.71 | 81.51±2.12 | 79.71±2.85 |
| **DROPS** | ✗ | ✓ | 90.67±0.76 | 89.05±0.88 | 86.80±1.34 | 85.89±1.55 | 84.67±1.88 | 82.59±2.50 | 79.44±3.58 |
| G-DRO | ✓ | ✓ | 92.59±0.87 | 90.95±0.52 | 90.18±0.46 | 89.89±0.43 | 89.53±0.37 | 88.97±0.26 | 88.21±0.17 |
| SUBG | ✓ | ✓ | 91.09±0.62 | 88.75±0.34 | 87.69±0.31 | 87.27±0.41 | 86.72±0.73 | 85.80±0.29 | 84.45±0.28 |
| $DFR_{Tr}^{Tr}$ | ✓ | ✓ | 90.36±0.64 | 89.25±0.73 | 87.30±0.97 | 86.53±1.09 | 85.50±1.26 | 83.77±1.63 | 81.22±2.31 |
| $DFR_{Tr}^{Val}$ | ✗ | ✓✓ | 90.90±0.79 | 91.13±0.40 | 90.32±0.54 | 90.02±0.58 | 89.64±0.65 | 89.05±0.77 | 88.25±1.03 |
| **DROPS*** | ✗ | ✓✓ | 95.69±0.41 | 93.59±0.36 | 92.64±0.46 | 92.28±0.51 | 91.82±0.58 | 91.10±0.68 | 90.19±0.81 |

## 7 CONCLUSIONS

We study the problem of improving the the distributional robustness of a pre-trained model under controlled distribution shifts. We propose DROPS, a fast and lightweight post-hoc method that learns scaling adjustments to predictions from a pre-trained model. DROPS learns the adjustments by solving a constrained optimization problem on a held-out validation set, and then applies these adjustments to the model predictions during evaluation time. DROPS is able to reuse the same pre-trained model for different robustness requirements by simply scaling the model predictions. For group robustness tasks, our approach only needs group annotations for a smaller held-out set. We also showed provable convergence guarantees for our method. Experimental results on standard benchmarks for class imbalance (CIFAR-10, CIFAR-100) and group DRO (Waterbirds, CelebA) demonstrate the effectiveness and robustness of DROPS when evaluated on a range of distribution shifts away from the target prior distribution.

ACKNOWLEDGEMENT

The work is done during JW's internship at Google Research, Brain Team. JW and YL are partially supported by the National Science Foundation (NSF) under grants IIS-2007951, IIS-2143895, and the Office of Naval Research under grant N00014-20-1-22. JW is also partially supported by the Center for Research in Open Source Software at UC Santa Cruz, which is funded by a donation from Sage Weil and industry memberships. We are thankful to Kevin Murphy for providing several helpful comments on the manuscript.

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

# A PROOFS

## A.1 PROOF OF THEOREM 1

The proof is adaptation of a similar result in Wang et al. (2022). We first prove Lemma 2.

*Proof of Lemma 2.* We first expand the objective:

$$\sum_{i=1}^{m} g_i \cdot \ell_i(f) = \sum_{i=1}^{m} g_i \cdot \mathbb{E}_{x|y=i} [\ell(i, f(x))] = \sum_{i=1}^{m} \frac{g_i}{\pi_i} \cdot \mathbb{E}_x [\eta_i(x) \cdot \ell(i, f(x))].$$

Given that $\ell$ is a proper loss, we have that the minimizer of this objective takes the form:

$$f_i(x) = \frac{\frac{g_i}{\pi_i} \cdot \eta_i(x)}{\sum_{j=1}^{m} \frac{g_j}{\pi_j} \cdot \eta_j(x)}.$$

$\square$

We are now ready to prove Theorem 1.

*Proof of Theorem 1.* The min-max problem in equation 3 can be expanded as:

$$\min_{f:\mathcal{X}\to\Delta_m} \mathrm{DRL}(f;\delta) = \min_{f:\mathcal{X}\to\Delta_m} \max_{g\in\mathbb{G}(\delta)} \underbrace{\sum_{y=1}^{m} \frac{g_y}{\pi_y} \cdot \mathbb{E}[\eta_y(x) \cdot \ell(y, f(x))]}_{\omega(g,f)}. \qquad (6)$$

Note that the objective $\omega(g, f)$ is clearly linear in $g$ (for fixed $f$), and with $\ell$ chosen to be convex in $f(x)$ (for fixed $g$), i.e., $\omega(g, \kappa f_1 + (1 - \kappa)f_2) \leq \kappa\omega(g, f_1) + (1 - \kappa)\omega(g, f_2), \forall f_1, f_2 : \mathcal{X} \to \Delta_m, \kappa \in [0, 1]$. Furthermore, given that divergence $D(g, \cdot)$ is convex in $g$, we have that $\mathbb{G}(\delta)$ is a convex compact set, while the domain of $f$ is convex. It follows from Sion's minimax theorem (Sion, 1958) that:

$$\min_{f:\mathcal{X}\to\Delta_m} \max_{g\in\mathbb{G}(\delta)} \omega(g, f) = \max_{g\in\mathbb{G}(\delta)} \min_{f:\mathcal{X}\to\Delta_m} \omega(g, f). \qquad (7)$$

Let $(g^*, f^*)$ be such that:

$$f^* \in \operatorname*{argmin}_{f:\mathcal{X}\to\Delta_m} \mathrm{DRL}(f;\delta) = \operatorname*{argmin}_{f:\mathcal{X}\to\Delta_m} \max_{g\in\mathbb{G}(\delta)} \omega(g, f);$$

$$g^* \in \operatorname*{argmax}_{g\in\mathbb{G}(\delta)} \min_{f:\mathcal{X}\to\Delta_m} \omega(g, f).$$

Such an $f^*$ exists because $\mathrm{DRL}(f;\delta)$ takes a bounded value when $f = \eta$, and any minimizer of $\mathrm{DRL}(f;\delta)$ yields a value below that; because $\mathrm{DRL}(f;\delta) \geq 0$ and is convex in $f$, such a minimizer exits. Similarly, $g^*$ also exists because $\mathbb{G}(\delta)$ is a compact set, and for any fixed $g$, $\min_{f:\mathcal{X}\to\Delta_m} \omega(g, f)$ is bounded above (owing to the existence of a minimizer from Lemma 2).

We now show that $(g^*, f^*)$ is a saddle-point for equation 6, i.e.,

$$\omega(g^*, f^*) = \max_{g\in\mathbb{G}(\delta)} \omega(g, f^*) = \min_{f:\mathcal{X}\to\Delta_m} \omega(g^*, f). \qquad (8)$$

To see this, notice that:

$$\omega(g^*, f^*) \leq \max_{g\in\mathbb{G}(\delta)} \omega(g, f^*)$$

$$= \min_{f:\mathcal{X}\to\Delta_m} \max_{g\in\mathbb{G}(\delta)} \omega(g, f) = \max_{g\in\mathbb{G}(\delta)} \min_{f:\mathcal{X}\to\Delta_m} \omega(g, f)$$

$$= \min_{f:\mathcal{X}\to\Delta_m} \omega(g^*, f) \leq \omega(g^*, f^*),$$

where we are able to swap the min and max in the second step using equation 7.

We thus from equation 8 that $f^*$ is a minimizer of $\omega(g^*, f)$, i.e.,

$$f^* \in \operatorname*{argmin}_{f:\mathcal{X}\to\Delta_m} \sum_{y=1}^{m} \frac{g_y^*}{\pi_y} \cdot \mathbb{E}\left[\eta_y(x) \cdot \ell(y, f(x))\right].$$

Following Lemma 2, we further have that for any $x \in \mathcal{X}$:

$$f^*(x) \propto \frac{g_y^*}{\pi_y} \eta_y(x).$$

$\square$

## A.2 Proof of Proposition 1

*Proof.* Now we prove the equivalence of two optimization tasks:

$$\min_{f:\mathcal{X}\to\Delta_m} \mathrm{DRL}(f; \delta) \quad \Longleftrightarrow \quad \min_{f:\mathcal{X}\to\Delta_m} \max_{g\in\Delta_m} \min_{\lambda\geq 0} \mathcal{L}(f, g, \lambda).$$

"$\Longrightarrow$" Remember that

$$\min_{f:\mathcal{X}\to\Delta_m} \mathrm{DRL}(f; \delta) = \min_{f:\mathcal{X}\to\Delta_m} \max_{g\in\mathbb{G}(\delta)} \sum_{i=1}^{m} g_i \cdot \ell_i(f),$$

we firstly show that any $(f^*, g^*)$ given by (L.H.S) yields the optimum of the equation 4 (R.H.S), for some $\lambda^*$. Note that for any $f, g \in \mathbb{G}(\delta)$, for any $g$ such that $D(g, r) \leq \delta$, we have:

$$\min_{f:\mathcal{X}\to\Delta_m} \max_{g\in\mathbb{G}(\delta)} \sum_{i\in[m]} g_i \ell_i(f) \geq \sum_{i\in[m]} g_i^* \ell_i(f^*).$$

Plugging $f^*, g^*$ into the R.H.S, we then have:

$$\mathcal{L}(f^*, g^*, \lambda) = \sum_{i\in[m]} g_i^* \ell_i(f^*) - \lambda\left(D(g^*, r) - \delta\right).$$

Since $D(g^*, r) \leq \delta$, $\exists \lambda^* \in \mathcal{R}_+$ such that the optimization task $\lambda^* = \arg\min_{\lambda\in\mathbb{R}_+} \mathcal{L}(f^*, g^*, \lambda)$. To show $(f^*, g^*, \lambda^*)$ returns the optimum of the R.H.S, we prove by contradiction and assume that there exists $f', g', \lambda'$ such that: $\mathcal{L}(f', g', \lambda) < \mathcal{L}(f^*, g^*, \lambda^*)$. This indicates that:

$$\left(\sum_i g_i' \ell_i(f') - \sum_i g_i^* \ell_i(f^*)\right) - \left(\lambda'\left(D(g', r) - \delta\right) - \lambda^*\left(D(g^*, r) - \delta\right)\right) < 0$$

$$\Longleftrightarrow \sum_i g_i' \ell_i(f') - \sum_i g_i^* \ell_i(f^*) < 0. \qquad \text{(Due to Complementary Slackness Condition)}$$

This contradicts with the fact that $\sum_i g_i' \ell_i(f') \geq \sum_i g_i^* \ell_i(f^*)$. Thus, $(f^*, g^*, \lambda^*)$ returns the optimum of R.H.S.

"$\Longleftarrow$"

We next prove that any $(f^*, g^*, \lambda^*)$ given by R.H.S yields the optimum in L.H.S. Due to Complementary Slackness Condition, we have: $\lambda^*\left(D(g^*, r) - \delta\right) = 0$. Note that $\lambda^* \in \mathcal{R}_+$, we then have $D(g^*, r) - \delta = 0$ and the constraint in L.H.S is satisfied. Thus, we have: R.H.S $= \sum_i g_i^* \ell_i(f^*)$. Again, if there exists $g', f'$ such that the L.H.S is minimized, where $g' \neq g^*, f' \neq f^*$, we then have $(g', f', \lambda' = 0)$ which satisfies: $\mathcal{L}(f', g', \lambda') < \mathcal{L}(f^*, g^*, \lambda^*)$, which contradicts with the argmin pairs $(f^*, g^*, \lambda^*)$.

Thus, we finished the proof. $\square$

## A.3 Proof of Theorem 3

The proof builds on prior convergence results in constrained and robust optimization (Narasimhan et al., 2019; Wang et al., 2022). We will find it useful to define the following quantities:

$$\mathcal{L}_1(f, g) = \sum_{i=1}^{m} g_i \cdot \ell_i(f); \quad \hat{\mathcal{L}}_1(f, g) = \sum_{i=1}^{m} g_i \cdot \hat{\ell}_i(f);$$

$$\mathcal{L}_2(g, \lambda) = -\lambda(D(g, u) - \delta).$$

We further define averages of different iterates $\bar{\lambda} = \frac{1}{T}\sum_{t=1}^{T} \lambda^{(t)}$ and $\bar{g} = \frac{1}{T}\sum_{t=1}^{T} g^{(t)}$.

We would also find the following lemmas useful. The first is a bound on our estimate of the class priors.

**Lemma 4.** *Under the assumptions in Theorem 1, with probability at least $1 - \alpha/2$ over draw of validation sample $S \sim \mathcal{D}^n$:*

$$\hat{\pi}_y \geq \frac{1}{2Z}, \forall y \in [m].$$

*Proof.* The proof follows from a direct application of Chernoff's bound (along with a union bound over all $m$ classes), noting that $\min_{y \in [m]} \pi_y \geq \frac{1}{Z}$ and $n \geq 8Z \log(2m/\alpha)$. □

Throughout the proof, we will assume that the statement in the above lemma holds with probability at least $1 - \alpha/2$.

Our second lemma shows that the equivalence between the saddle-point optimization in equation 4 and the original constrained optimization problem in equation 3 still holds when we minimize the Lagrange multiplier only over a bounded set:

**Lemma 5.** *Under the assumptions in Theorem 1, we have for any $f' : \mathcal{X} \to \Delta_m$:*

$$\min_{\lambda \in [0, R]} \max_{g \in \Delta_m} \mathcal{L}(f', g, \lambda) = \max_{g \in \mathbb{G}(\delta)} \sum_{i=1}^{m} g_i \cdot \ell_i(f').$$

*Proof.* Let $\lambda^* \in \operatorname*{argmin}_{\lambda \geq 0} \max_{g \in \Delta_m} \mathcal{L}(f', g, \lambda)$ be the $\lambda$-minimizer over all non-negative $\mathbb{R}$. Such a minimizer exists for the following reason. Owing to the continuity of $D$ we know there exits at least one $g' \in \Delta_m$ for which $D(g, u) = \delta$ and therefore we have that the minimization objective is bounded:

$$\max_{g \in \Delta_m} \mathcal{L}(f', g, \lambda) \leq \mathcal{L}(f', g', \lambda) = \sum_{i=1}^{m} g_i \cdot \ell_i(f') \leq Z B_\ell.$$

It remains to be shown that $\lambda^* \leq R$. To do this end, let

$$g^* \in \operatorname*{argmax}_{g \in \Delta_m : D(g, u) \leq \delta} \sum_{i=1}^{m} g_i \cdot \ell_i(f').$$

We note that:

$$\sum_{i=1}^{m} g_i^* \cdot \ell_i(f') = \min_{\lambda \geq 0} \max_{g \in \Delta_m} \mathcal{L}(f', g, \lambda) = \max_{g \in \Delta_m} \mathcal{L}(f', g, \lambda^*) = \max_{g \in \Delta_m} \sum_{i=1}^{m} g_i \cdot \ell_i(f') - \lambda^*(D(g, u) - \delta).$$

Choose $\tilde{g}$ such that $D(\tilde{g}, u) = \delta/2$, which exits thanks to the continuity of $D$. Upper bounding the max on RHS in the above equality by substituting $\tilde{g}$, we get:

$$\sum_{i=1}^{m} g_i^* \cdot \ell_i(f') \leq \sum_{i=1}^{m} g_i \cdot \ell_i(f') - \lambda^*(D(\tilde{g}, u) - \delta) = \sum_{i=1}^{m} g_i \cdot \ell_i(f') - \lambda^* \delta/2,$$

which gives us:

$$\lambda^* \leq \frac{2}{\delta}\left(\sum_{i=1}^{m} g_i \cdot \ell_i(f') - \sum_{i=1}^{m} g_i^* \cdot \ell_i(f')\right) \leq \frac{2}{\delta} Z B_\ell = R,$$

as desired. □

The lemmas below follow from Lemma 4 and standard results in online convex optimization (Shalev-Shwartz et al., 2011).

**Lemma 6.** *Under the assumptions in Theorem 1, and for $\eta_g = \frac{1}{2B_\ell Z + RL_D}\sqrt{\frac{\log(m)}{T}}$, with probability at least $1 - \alpha/2$ over draw of $S \sim \mathcal{D}^n$, the sequence of iterates $g^{(1)}, \ldots, g^{(T)}$ satisfies:*

$$\max_{g \in \Delta_m} \frac{1}{T}\sum_{t=1}^{T} \hat{\mathcal{L}}_1(f^{(t)}, g, \lambda^{(t)}) - \frac{1}{T}\sum_{t=1}^{T} \hat{\mathcal{L}}_1(f^{(t)}, g^{(t)}, \lambda^{(t)}) \leq 2(2B_\ell Z + RL_D)\sqrt{\frac{\log(m)}{T}}.$$

*Proof.* The proof follows from standard convergence result for the exponentiated-gradient descent algorithm noting that

$$\|\nabla_g \hat{\mathcal{L}}(f^{(t)}, g^{(t)}, \lambda^{(t)})\|_\infty \leq \max_i |\hat{\ell}_i(f^{(t)})| + |\lambda^{(t)}| \|\nabla_g D(g^{(t)}, u)\|_\infty$$

$$\leq B_\ell \cdot \max_i \frac{1}{\hat{\pi}_i} + RL_D \leq 2ZB_\ell + RL_D,$$

where we use the bound on the class prior estimates in Lemma 4. The last step holds with probability at least $1 - \alpha/2$. $\qquad\square$

**Lemma 7.** *Under the assumptions in Theorem 1, and for $\eta_\lambda = \frac{R}{B_D\sqrt{T}}$ with probability at least $1 - \alpha/2$ over draw of validation sample $S \sim \mathcal{D}^n$, the sequence of iterates $\lambda^{(1)}, \ldots, \lambda^{(T)}$ satisfies:*

$$\frac{1}{T}\sum_{t=1}^{T} \mathcal{L}_2(g^{(t)}, \lambda^{(t)}) - \min_{\lambda \in [0,R]} \frac{1}{T}\sum_{t=1}^{T} \mathcal{L}_2(g^{(t)}, \lambda) \leq \frac{RB_D}{\sqrt{T}}.$$

*Proof.* The proof follows from standard convergence result for the online gradient descent algorithm noting that $|\nabla_\lambda \mathcal{L}_2(g^{(t)}, \lambda)| \leq |D(g^{(t)}, u) - \delta| \leq B_D$ and $|\lambda| \leq R$. $\qquad\square$

The following lemma provides a generalization bound for the empirical Lagrangian.

**Lemma 8.** *Under the assumptions in Theorem 1, with probability at least $1 - \alpha$ over draw of validation sample $S \sim \mathcal{D}^n$, for all $t \in [T]$:*

$$|\mathcal{L}(f^{(t)}, g^{(t)}, \lambda^{(t)}) - \hat{\mathcal{L}}(f^{(t)}, g^{(t)}, \lambda^{(t)})| \leq \mathcal{O}\left(\sqrt{\frac{\log(m|\mathcal{F}|/\alpha)}{n}}\right).$$

*Proof.* For any $t \in [T]$, we first bound the left-hand side by:

$$|\mathcal{L}(f^{(t)}, g^{(t)}, \lambda^{(t)}) - \hat{\mathcal{L}}(f^{(t)}, g^{(t)}, \lambda^{(t)})| \leq \sum_{i=1}^{m} g_i^{(t)} \cdot \left|\ell_i(f) - \hat{\ell}_i(f)\right| \leq \max_{i \in [m]}\left|\ell_i(f) - \hat{\ell}_i(f)\right|. \quad (9)$$

Further define $\tilde{\ell}_i = \frac{1}{\pi_i}\sum_{(x,y) \in S: y = i} \ell(y, f(x))$. We then can bound the above difference $\left|\ell_i(f) - \hat{\ell}_i(f)\right|$ in the above bound using:

$|\ell_i(f) - \hat{\ell}_i(f)|$

$\leq |\ell_i(f) - \tilde{\ell}_i(f)| + |\tilde{\ell}_i(f) - \hat{\ell}_i(f)|$

$= \frac{1}{\pi_i}\left|\mathbb{E}\left[\ell(i, f(x)) \cdot \mathbf{1}(y = i)\right] - \frac{1}{n}\sum_{j=1}^{n} \ell(i, f(x_j)) \cdot \mathbf{1}(y_j = i)\right| + \left|\frac{1}{\pi_i} - \frac{1}{\hat{\pi}_i}\right|\frac{1}{n}\sum_{j=1}^{n} \ell(i, f(x_j))\,\mathbf{1}(y_j = i)$

$\leq Z\left|\mathbb{E}\left[\ell(i, f(x)) \cdot \mathbf{1}(y = i)\right] - \frac{1}{n}\sum_{j=1}^{n} \ell(i, f(x_j)) \cdot \mathbf{1}(y_j = i)\right| + \frac{B_\ell}{\pi_i \hat{\pi}_i}|\pi_i - \hat{\pi}_i|$

We know $\pi_i \leq \frac{1}{Z}$. Further, applying Lemma 4, we can bound $\hat{\pi}_i$. We therefore have with probability at least $1 - \alpha/2$ over draw of $S \sim \mathcal{D}^n$:

$$|\ell_i(f) - \hat{\ell}_i(f)| \leq Z \left| \mathbb{E}\left[\ell(i, f(x)) \cdot \mathbf{1}(y = i)\right] - \frac{1}{n} \sum_{j=1}^{n} \ell(i, f(x_j)) \cdot \mathbf{1}(y_j = i) \right| + 2Z^2 B_\ell \left|\pi_i - \hat{\pi}_i\right|.$$

An application of Hoeffding's inequality to both the above terms, noting that the loss $\ell(y, z)$ is bonded, together with a union bound over all $f \in \mathcal{F}$ and class $i \in [m]$, gives us with probability at least $1 - \alpha/2$ over draw of $S \sim \mathcal{D}^n$, for all $f \in \mathcal{F}$ and $i \in [m]$:

$$|\ell_i(f) - \hat{\ell}_i(f)| \leq \mathcal{O}\left( \sqrt{\frac{\log(m|\mathcal{F}|/\alpha)}{n}} \right).$$

Plugging back into equation 9 and taking a union bound over both the high probability statements completes the proof. $\qquad \square$

We will additionally use the following regret bound for the $f$-minimization step:

**Lemma 9.** *Under the assumptions in Theorem 1, for a fixed $g \in \Delta_m$, with probability at least $1 - \alpha$ over draw of $S \sim \mathcal{D}^n$,*

$$\mathcal{L}_1(f^{(t)}, g) - \min_{f \in \mathcal{F}} \mathcal{L}_1(f, g) \leq B_\ell Z \cdot \mathbb{E}_x\left[\|\hat{\eta}(x) - \eta(x)\|_1\right] + \mathcal{O}\left( \sqrt{\frac{\log(m/\alpha)}{n}} \right)$$

*Proof.* We first expand $\mathcal{L}_1$ in terms of the conditional-class probability function $\eta(x)$:

$$\mathcal{L}_1(f, g) = \sum_{i=1}^{n} g_i \cdot \ell_i(f) = \sum_{i=1}^{n} \frac{g_i}{\pi_i} \cdot \mathbb{E}_x\left[\eta(x) \cdot \ell(i, f(x))\right].$$

We know from Lemma 2 that the minimizer of $\mathcal{L}_1(f, g)$ over all $f$ takes the form $f^*(x) \propto \frac{g_y^*}{\pi_y} \eta_y(x)$. We also know from Lemma 2 that $f^{(t)}$ is the minimizer of a similar objective where $\eta$ is replaced by the pre-trained model $\hat{\eta}$:

$$\tilde{\mathcal{L}}_1(f, g) = \sum_{i=1}^{n} \frac{g_i}{\hat{\pi}_i} \cdot \mathbb{E}_x\left[\hat{\eta}(x) \cdot \ell(i, f(x))\right].$$

We would like to bound:

$$\begin{aligned}
\mathcal{L}_1(f^{(t)}, g) - \mathcal{L}_1(f^*, g) &= \mathcal{L}_1(f^{(t)}, g) - \tilde{\mathcal{L}}_1(f^{(t)}, g) + \tilde{\mathcal{L}}_1(f^{(t)}, g) - \mathcal{L}_1(f^*, g) \\
&\leq \mathcal{L}_1(f^{(t)}, g) - \tilde{\mathcal{L}}_1(f^{(t)}, g) + \tilde{\mathcal{L}}_1(f^*, g) - \mathcal{L}_1(f^*, g) \\
&= \mathbb{E}_x\left[\sum_{i=1}^{n} g_i \cdot \left(\frac{\eta_i(x)}{\pi_i} - \frac{\hat{\eta}_i(x)}{\hat{\pi}_i}\right) \cdot \ell(i, f^{(t)}(x))\right] + \mathbb{E}_x\left[\sum_{i=1}^{n} g_i \cdot \left(\frac{\eta_i(x)}{\pi_i} - \frac{\hat{\eta}_i(x)}{\hat{\pi}_i}\right) \cdot \ell(i, f^*(x))\right] \\
&= \mathbb{E}_x\left[\sum_{i=1}^{n} g_i \cdot \left(\ell(i, f^{(t)}(x)) - \ell(i, f^*(x))\right) \cdot \left(\frac{\eta_i(x)}{\pi_i} - \frac{\hat{\eta}_i(x)}{\hat{\pi}_i}\right)\right] \\
&\leq \mathbb{E}_x\left[\max_{i \in [m]} g_i \cdot \left|\ell(i, f^{(t)}(x)) - \ell(i, f^*(x))\right| \cdot \sum_{i=1}^{m} \left|\frac{\eta_i(x)}{\pi_i} - \frac{\hat{\eta}_i(x)}{\hat{\pi}_i}\right|\right] \\
&\leq B_\ell \cdot \mathbb{E}_x\left[\sum_{i=1}^{m} \left|\frac{\eta_i(x)}{\pi_i} - \frac{\hat{\eta}_i(x)}{\hat{\pi}_i}\right|\right] \\
&= B_\ell \cdot \mathbb{E}_x\left[\sum_{i=1}^{m} \left|\frac{\eta_i(x)}{\pi_i} - \frac{\hat{\eta}_i(x)}{\pi_i} + \frac{\hat{\eta}_i(x)}{\pi_i} - \frac{\hat{\eta}_i(x)}{\hat{\pi}_i}\right|\right]
\end{aligned}$$

$$\leq B_\ell \cdot \max_{i \in [m]} \frac{1}{\pi_i} \cdot \mathbb{E}_x\left[\|\eta(x) - \hat\eta(x)\|\right] + \mathbb{E}_x\left[\sum_{i=1}^m \left|\frac{\hat\eta_i(x)}{\pi_i} - \frac{\hat\eta_i(x)}{\hat\pi_i}\right|\right]$$

$$\leq B_\ell Z \cdot \mathbb{E}_x\left[\|\eta_i(x) - \hat\eta_i(x)\|_1\right] + \mathbb{E}_x\left[\|\hat\eta(x)\|\right]_1 \cdot \max_{i \in [m]} \left|\frac{1}{\pi_i} - \frac{1}{\hat\pi_i}\right|$$

$$= B_\ell Z \cdot \mathbb{E}_x\left[\|\eta_i(x) - \hat\eta_i(x)\|_1\right] + (1) \cdot \max_{i \in [m]} \frac{1}{\pi_i \hat\pi_i} |\pi_i - \hat\pi_i|,$$

where in the second step, we use the fact that $f^{(t)}$ minimizes $\tilde{\mathcal{L}}_1$; in the second-last step, we apply Holder's inequality; in the last step, we use the fact that $g_i \in [0, 1]$, $\pi_i \leq \frac{1}{Z}$ and $\ell(y, z) \leq B_\ell$.

We know $\pi_i \leq \frac{1}{Z}$. Further, applying Lemma 4, we can bound $\hat\pi_i$. We have with probability at least $1 - \alpha/2$ over draw of $S \sim \mathcal{D}^n$:

$$\mathcal{L}_1(f^{(t)}, g) - \mathcal{L}_1(f^*, g) \leq B_\ell Z \cdot \mathbb{E}_x\left[\|\eta_i(x) - \hat\eta_i(x)\|_1\right] + 2Z^2 \cdot \max_{i \in [m]} |\pi_i - \hat\pi_i|.$$

An application of Hoeffding's inequality to the last term completes the proof. $\qquad\square$

We are now ready to prove Theorem 3.

*Proof of Theorem 3.* Let $\kappa_n = \mathcal{O}\left(\sqrt{\frac{\log(m|\mathcal{F}|/\alpha)}{n}} + \mathbb{E}_x\left[\|\hat\eta(x) - \eta(x)\|_1\right]\right)$. We start by combining Lemma 6 with the generalization bound in Lemma 8, from which we have with probability at least $1 - \alpha$ over draw of validation sample $S \sim \mathcal{D}^n$,

$$\max_{g \in \Delta_m} \frac{1}{T} \sum_{t=1}^T \mathcal{L}(f^{(t)}, g, \lambda^{(t)}) \leq \frac{1}{T} \sum_{t=1}^T \mathcal{L}(f^{(t)}, g^{(t)}, \lambda^{(t)}) + \mathcal{O}\left(\sqrt{\frac{\log(m)}{T}} + \sqrt{\frac{\log(m|\mathcal{F}|/\alpha)}{n}}\right)$$

$$= \frac{1}{T} \sum_{t=1}^T \mathcal{L}_1(f^{(t)}, g^{(t)}) + \frac{1}{T} \sum_{t=1}^T \mathcal{L}_2(g^{(t)}, \lambda^{(t)}) + \mathcal{O}\left(\sqrt{\frac{\log(m)}{T}} + \sqrt{\frac{\log(m|\mathcal{F}|/\alpha)}{n}}\right)$$

$$= \frac{1}{T} \sum_{t=1}^T \mathcal{L}_1(f^{(t)}, g^{(t)}) + \frac{1}{T} \sum_{t=1}^T \mathcal{L}_2(g^{(t)}, \lambda^{(t)}) + \mathcal{O}\left(\sqrt{\frac{\log(m|\mathcal{F}|/\alpha)}{n}}\right),$$

$$\tag{10}$$

where we have used the fact that $T = O(n)$.

Applying Lemma 7 to the right-hand side of equation 10, with $T = O(n)$, we have with probability at least $1 - \alpha$,

$$\max_{g \in \Delta_m} \frac{1}{T} \sum_{t=1}^T \mathcal{L}(f^{(t)}, g, \lambda^{(t)}) \leq \frac{1}{T} \sum_{t=1}^T \mathcal{L}_1(f^{(t)}, g^{(t)}) + \min_{\lambda \in [0,R]} \frac{1}{T} \sum_{t=1}^T \mathcal{L}_2(g^{(t)}, \lambda) + \mathcal{O}\left(\sqrt{\frac{\log(m|\mathcal{F}|/\alpha)}{n}}\right)$$

$$\leq \min_{f:\mathcal{X} \to \Delta_m} \frac{1}{T} \sum_{t=1}^T \mathcal{L}_1(f, g^{(t)}) + \min_{\lambda \in [0,R]} \frac{1}{T} \sum_{t=1}^T \mathcal{L}_2(g^{(t)}, \lambda) + \kappa_n$$

$$\leq \min_{f:\mathcal{X} \to \Delta_m} \mathcal{L}_1(f, \bar{g}) + \min_{\lambda \in [0,R]} \mathcal{L}_2(\bar{g}, \lambda) + \kappa_n, \tag{11}$$

where in the pen-ultimate step, we apply Lemma 9, and the last step uses the fact that $\mathcal{L}_1(f, g)$ is linear in $g$ and applies Jensen's inequality to $\mathcal{L}_2(f, g)$ noting that is concave in $g$ (as a result of $-D(g, u)$ being concave in $g$).

Applying Jensen's inequality again to the LHS of equation 11, noting that $\ell_i(f) = \mathbb{E}\left[\ell(y, f(x) \mid y = i\right]$ and as a result $\mathcal{L}_1(f, g)$ is convex in $f$ (owing to $\ell(y, z)$ being convex in $z$) and additionally using the fact that $\mathcal{L}_2(g, \lambda)$ is linear in $\lambda$, we further have:

$$\max_{g \in \Delta_m} \mathcal{L}(\bar{f}, g, \bar\lambda) \leq \min_{f:\mathcal{X} \to \Delta_m, \lambda \in [0,R]} \mathcal{L}(f, \bar{g}, \lambda) + \kappa_n.$$

Lower bounding the LHS by a min over $\lambda \in [0, R]$ (noting that $\bar{\lambda} \in [0, R]$), and the RHS by a max over $g \in \Delta_m$, we have:

$$\min_{\lambda \in [0,R]} \max_{g \in \Delta_m} \mathcal{L}(\bar{f}, g, \bar{\lambda}) \leq \max_{g \in \Delta_m} \min_{f:\mathcal{X} \to \Delta_m, \lambda \in [0,R]} \mathcal{L}(f, \bar{g}, \lambda) + \kappa_n.$$

Exchanging the min's and max's using min-max theorem,

$$\max_{g \in \Delta_m} \min_{\lambda \in [0,R]} \mathcal{L}(\bar{f}, g, \lambda) \leq \min_{f:\mathcal{X} \to \Delta_m} \max_{g \in \Delta_m} \min_{\lambda \in [0,R]} \mathcal{L}(f, g, \lambda) + \kappa_n.$$

In other words for any $f^* : \mathcal{X} \to \Delta_m$,

$$\max_{g \in \Delta_m} \min_{\lambda \in [0,R]} \mathcal{L}(\bar{f}, g, \lambda) \leq \max_{g \in \Delta_m} \min_{\lambda \in [0,R]} \mathcal{L}(f^*, g, \lambda) + \kappa_n.$$

An application of Lemma 5 to both the LHS and RHS gives us for any $f^* : \mathcal{X} \to \Delta_m$,

$$\max_{g \in \mathbb{G}(\delta)} \sum_{i=1}^{m} g_i \cdot \ell_i(\bar{f}) \leq \max_{g \in \mathbb{G}(\delta)} \sum_{i=1}^{m} g_i \cdot \ell_i(f^*) + \kappa_n,$$

which completes the proof.

$\square$

## A.4 EG-UPDATE FOR $g^{(t)}$

To obtain the updates for $g$, we consider the fixed $\lambda^{(t)} \in \mathbb{R}_+$ and adopt a EG-style computation. Taking the KL divergence for illustration, we provide the closed form of $g^{(t+1)}$ in Proposition 2.

**Proposition 2.** *(Un-normalized) EG-updates for $g^{(t)}$ under $D_{KL}$ is given by:*

$$g_i^{(t+1)} = \left( g_i^{(t)} \exp\{\eta_g \ell_i(f) + \lambda \eta_g \log(u_i)\} \right)^{\frac{1}{1+\lambda \eta_g}} .$$

Regarding the EG-updates for $g^{(t)}$, it is straightforward from Proposition 2 that classes with a larger loss or a higher target distribution weight $r_i$ tend to receive a larger weight.

*Proof.* In this proof, we consider the generic target distribution $r$ which covers the uniform prior $u$ as a special case. To obtain $g_i^{(t+1)}$ for KL divergence, where $D_{KL}(g, r) = \sum_i g_i \log \left( \frac{g_i}{r_i} \right)$, we need:

$$\frac{\partial f \left( -\frac{1}{\eta_g} \sum_i g_i \log \left( \frac{g_i}{g_i^{(t)}} \right) + \sum_i g_i \ell_i(f) - \lambda \left( \sum_i g_i \log \left( \frac{g_i}{r_i} \right) - \delta \right) \right)}{\partial g_i} = 0$$

$$\implies \frac{-1}{\eta_g} \log \left( \frac{g_i}{g_i^{(t)}} \right) - \frac{1}{\eta_g} g_i \left( \frac{g_i^{(t)}}{g_i} \right) \left( \frac{1}{g_i^{(t)}} \right) + \ell_i(f) - \lambda \log \left( \frac{g_i}{r_i} \right) - \lambda g_i \left( \frac{r_i}{g_i} \right) \left( \frac{1}{r_i} \right) = 0$$

$$\implies \frac{-1}{\eta_g} \log \left( \frac{g_i}{g_i^{(t)}} \right) - \frac{1}{\eta_g} + \ell_i(f) - \lambda \log \left( \frac{g_i}{r_i} \right) - \lambda = 0$$

$$\implies \ell_i(f) - \frac{1}{\eta_g} - \lambda = \lambda \log \left( \frac{g_i}{r_i} \right) + \frac{1}{\eta_g} \log \left( \frac{g_i}{g_i^{(t)}} \right)$$

$$\implies \ell_i(f) - \frac{1}{\eta_g} - \lambda + \lambda \log(r_i) + \frac{1}{\eta_g} \log(g_i^{(t)}) = \lambda \log(g_i) + \frac{1}{\eta_g} \log(g_i)$$

$$\implies (\lambda \eta_g + 1) \log(g_i) = \ell_i(f) \eta_g - 1 - \lambda \eta_g + \lambda \eta_g \log(r_i) + \log(g_i^{(t)})$$

$$\implies g_i = \exp \left\{ \frac{\ell_i(f) \eta_g - 1 - \lambda \eta_g + \lambda \eta_g \log(r_i) + \log(g_i^{(t)})}{\lambda \eta_g + 1} \right\}$$

$$\implies g_i = \exp \left\{ \frac{\log(g_i^{(t)}) + \ell_i(f) \eta_g + \lambda \eta_g \log(r_i)}{1 + \lambda \eta_g} - 1 \right\},$$

Remove the constant, we then have:

$$g_i = \exp\left\{\frac{\log(g_i^{(t)}) + \ell_i(f)\eta_g + \lambda\eta_g \log(r_i)}{1 + \lambda\eta_g}\right\}$$

$$\Longrightarrow g_i = \left(g_i^{(t)} \exp\{\eta_g \ell_i(f) + \lambda\eta_g \log(r_i)\}\right)^{\frac{1}{1+\lambda\eta_g}}.$$

$\square$

## B    EXTENSION TO GROUP-PRIOR SHIFTS

We now show how our theoretical results extend to the group-prior shift setting. Suppose each instance $x \in \mathcal{X}$ is associated with a group $a \in A$, $m = |Y| \times |A|$. We define the group-specific conditional-class probability to be $\eta_i(x, a) = \mathbb{P}(y = i|x, a)$ and the group-specific class priors $\pi_{a,i} = \mathbb{P}(y = i|a)$. In this case, we wish to learn a scoring function $f : \mathcal{X} \times A \to \Delta_m$ that takes both the instance $x$ and group $g$ into account. We use $\ell_{a,y}(f) = \mathbb{E}\left[\ell(y, f(x, a)|x, a\right]$ to denote the class-$i$ loss conditioned on group $g$. Our goal is to minimize the following group-specific DRE objective:

$$\text{DRL}(f; \delta) = \min_{f:\mathcal{X}\to\Delta_m} \max_{w\in\mathbb{G}(\delta)} \sum_{a,i} g_{a,i} \cdot \ell_{a,i}(f), \tag{12}$$

where $\mathbb{G}(\delta) = \{g \in \Delta_{m\times k} \,|\, D(g, r) \le \delta\}$ for some $\delta > 0$, divergence function $D : \Delta_m \times \Delta_m \to \mathbb{R}$, and target distribution $r \in \Delta_m$.

**Theorem 10** (Bayes-optimal scorer for group-prior shift). *Suppose $\ell(y, z)$ is a proper loss that is convex in its second argument and $D(g, \cdot)$ is convex in $g$. Let $\delta > 0$ be such that $\mathbb{G}(\delta)$ is non-empty. For some $g^* \in \mathbb{G}(\delta)$, then the optimal solution to equation 12 takes the form:*

$$f_y^*(x, a) \propto \frac{g_{a,y}^*}{\pi_{a,y}} \cdot \eta_y(x, a).$$

The proof follows the same steps as Theorem 1, except that it uses the following lemma instead of Lemma 2:

**Lemma 11.** *Suppose $\ell(y, z)$ is a proper loss. For any fixed $g \in \mathbb{R}_+^{k\times m}$, the following is a minimizer to the objective $\sum_{a,y} g_{a,y} \cdot \ell_{a,y}(f)$ over all measurable functions $f : \mathcal{X} \times A \to \Delta_m$:*

$$f_y^*(x, a) \propto \frac{g_{a,y}}{\pi_{a,y}} \cdot \eta_y(x, a).$$

*Proof.* We first expand the objective:

$$\sum_{a,i} g_{a,i} \cdot \ell_{a,i}(f) = \sum_{a,i} g_{a,i} \cdot \mathbb{E}_{x|a,y=i}\left[\ell(i, f(x, a))\right] = \sum_{a,i} \frac{g_{a,i}}{\pi_{a,i}} \cdot \mathbb{E}_x\left[\eta_i(x, a) \cdot \ell(i, f(x, a))\right].$$

Given that $\ell$ is a proper loss, we have that the minimizer of this objective takes the form:

$$f_i^*(x, a) = \frac{\frac{g_{a,i}}{\pi_{a,i}} \cdot \eta_i(x, a)}{\sum_j \frac{g_{a,j}}{\pi_{a,j}} \cdot \eta_j(x, a)}.$$

$\square$

## C    ADDITIONAL EXPERIMENT DETAILS AND RESULTS

In this section, we introduce omitted experiment details and additional experiment results of our proposed methods.

## C.1 ABLATION STUDY OF DROPS ON CLASS-IMBALANCED CIFAR

We offer the ablation study of DROPS in Table 3. Suppose the designer is interested in $\delta_{\text{eval}} = 1.0$ robustness, setting $\delta_{\text{train}} \in [0.5, 1.0]$ frequently reaches best three performances in mean/($\delta_{\text{eval}} = 1.0$)-worst/worst by referring to the performance of averaged 5 runs, which indicates that DROPS is less sensitive to the parameter $\delta_{\text{train}}$. Setting $\delta_{\text{train}}$ to be a coarse estimate of the $\delta_{\text{eval}}$ should be able to appropriately improve the model robustness under prior shifts.

Table 3: Ablation study of DROPS on class-imbalanced CIFAR-100 dataset: mean ± std of averaged class accuracy, $\delta = 1.0$-worst case accuracy, and worst class accuracy of 5 runs are reported. The best three performed $\delta$ for in each setting are highlighted.

| Method | CIFAR-100 (Averaged) | | |
|---|---|---|---|
| | $\rho = 10$ | $\rho = 50$ | $\rho = 100$ |
| DROPS ($\delta = 0.1$) | 59.08±0.38 | 48.24±0.63 | 43.37±0.62 |
| DROPS ($\delta = 0.2$) | 59.12±0.21 | 47.90±0.52 | 43.58±0.61 |
| DROPS ($\delta = 0.3$) | 58.62±0.56 | 48.17±0.91 | 42.85±0.86 |
| DROPS ($\delta = 0.4$) | 58.98±0.37 | 47.64±0.88 | 42.60±1.15 |
| DROPS ($\delta = 0.5$) | 58.79±0.21 | 47.84±0.35 | 42.52±0.43 |
| DROPS ($\delta = 0.6$) | 59.35±0.27 | 47.44±0.65 | 42.47±0.84 |
| DROPS ($\delta = 0.7$) | 59.42±0.30 | 48.04±0.47 | 43.20±0.92 |
| DROPS ($\delta = 0.8$) | 59.63±0.10 | 47.78±0.96 | 43.44±0.60 |
| DROPS ($\delta = 0.9$) | 59.69±0.39 | 47.83±0.80 | 43.20±0.78 |
| DROPS ($\delta = 1.0$) | 59.60±0.63 | 47.96±0.86 | 43.23±1.45 |
| DROPS ($\delta = 1.1$) | 59.00±0.56 | 48.31±0.46 | 43.72±0.55 |
| DROPS ($\delta = 1.2$) | 59.36±0.37 | 48.02±1.30 | 43.11±0.35 |
| DROPS ($\delta = 1.3$) | 59.54±0.75 | 47.86±1.00 | 42.80±0.67 |

| Method | CIFAR-100 ($\delta = 1.0$-worst case acc) | | |
|---|---|---|---|
| | $\rho = 10$ | $\rho = 50$ | $\rho = 100$ |
| DROPS ($\delta = 0.1$) | 42.98±0.39 | 29.62±0.52 | 24.26±0.68 |
| DROPS ($\delta = 0.2$) | 43.96±0.18 | 29.98±1.00 | 25.16±0.49 |
| DROPS ($\delta = 0.3$) | 43.54±0.43 | 30.18±1.02 | 25.14±0.77 |
| DROPS ($\delta = 0.4$) | 44.16±0.76 | 30.50±0.96 | 24.90±0.88 |
| DROPS ($\delta = 0.5$) | 44.18±0.34 | 31.10±0.35 | 25.08±0.36 |
| DROPS ($\delta = 0.6$) | 44.64±0.35 | 30.12±0.73 | 24.60±0.64 |
| DROPS ($\delta = 0.7$) | 44.34±0.90 | 30.30±0.41 | 25.30±0.85 |
| DROPS ($\delta = 0.8$) | 44.16±0.43 | 30.32±0.68 | 25.48±0.55 |
| DROPS ($\delta = 0.9$) | 44.96±0.52 | 30.12±0.66 | 25.58±0.50 |
| DROPS ($\delta = 1.0$) | 44.86±1.05 | 31.14±0.74 | 26.24±1.88 |
| DROPS ($\delta = 1.1$) | 43.74±1.33 | 31.18±0.56 | 25.84±0.53 |
| DROPS ($\delta = 1.2$) | 44.92±0.69 | 31.28±1.27 | 25.06±0.58 |
| DROPS ($\delta = 1.3$) | 45.04±0.84 | 31.22±0.98 | 25.22±0.64 |

| Method | CIFAR-100 (Worst) | | |
|---|---|---|---|
| | $\rho = 10$ | $\rho = 50$ | $\rho = 100$ |
| DROPS ($\delta = 0.1$) | 20.61±3.24 | 8.25±2.46 | 6.02±1.50 |
| DROPS ($\delta = 0.2$) | 22.51±3.10 | 9.58±3.34 | 7.40±1.04 |
| DROPS ($\delta = 0.3$) | 23.32±4.39 | 10.90±3.90 | 7.29±1.90 |
| DROPS ($\delta = 0.4$) | 22.51±2.87 | 10.79±2.07 | 8.28±1.98 |
| DROPS ($\delta = 0.5$) | 25.13±3.05 | 12.52±2.57 | 6.70±2.39 |
| DROPS ($\delta = 0.6$) | 24.86±4.18 | 10.33±1.92 | 7.43±2.78 |
| DROPS ($\delta = 0.7$) | 21.74±4.97 | 10.89±2.07 | 9.03±2.67 |
| DROPS ($\delta = 0.8$) | 24.78±4.84 | 11.17±1.54 | 8.21±2.06 |
| DROPS ($\delta = 0.9$) | 25.98±3.95 | 11.65±3.05 | 8.61±2.22 |
| DROPS ($\delta = 1.0$) | 24.80±2.85 | 11.86±2.85 | 7.52±2.11 |
| DROPS ($\delta = 1.1$) | 24.22±4.83 | 12.81±2.15 | 9.42±1.99 |
| DROPS ($\delta = 1.2$) | 25.77±4.58 | 12.89±2.46 | 7.27±2.32 |
| DROPS ($\delta = 1.3$) | 26.49±3.66 | 12.54±1.41 | 8.56±1.02 |

## C.2 HYPOTHESIS TESTING OF PERFORMANCE COMPARISONS ON CLASS-IMBALANCED CIFAR

We provide the statistical testing of results in Table 1: in Table 4, we included the paired student t-test results between each baseline method and DROPS, for each dataset and each metric (mean/$\delta$-worst/worst accuracy), and the inputs of samples of each method for testing are the test accuracies

of 5 runs $\times$ 3 imbalance ratio settings). And each cell indicates the ($t$-statistics and $p$-value). It is quite obvious that for most results, there exists negative statistics, meaning that the given baseline method is significantly (if $p$-value is small enough, i.e., $p < 0.05$) worse than DROPS.

Table 4: Paired student t-test of the performance comparisons between each baseline method and DROPS: cells in right 6 columns denote (statistics, $p$-value) of the hypothesis testing results between each baseline method and DROPS, the scenario where negative statistics and $p$-value less than 0.05 indicates that DROPS is statistically significant better than the corresponding baseline method.

| Method V.S. DROPS | CIFAR-10 | | | CIFAR-100 | | |
|---|---|---|---|---|---|---|
| | Mean | $\delta = 1.0$-worst | Worst | Mean | $\delta = 1.0$-worst | Worst |
| Cross Entropy | $-11.3, 1.9e^{-8}$ | $-11.1, 2.6e^{-8}$ | $-9.7, 1.4e^{-7}$ | $-7.9, 1.5e^{-6}$ | $-12.9, 3.7e^{-7}$ | $-13.0, 3.3e^{-7}$ |
| Focal | $-4.9, 2.2e^{-4}$ | $-15.1, 4.9e^{-10}$ | $-21.2, 4.8e^{-12}$ | $-9.4, 3.6e^{-7}$ | $5.6, 8.8e^{-5}$ | $-4.6, 5.0e^{-4}$ |
| LDAM | $-9.1, 2.8e^{-7}$ | $-9.8, 1.2e^{-7}$ | $-9.2, 2.6e^{-7}$ | $-7.4, 3.1e^{-6}$ | $-6.6, 1.2e^{-5}$ | $-7.4, 3.2e^{-6}$ |
| Balanced-Softmax | $-4.1, 1.1e^{-3}$ | $-3.9, 1.6e^{-3}$ | $-3.6, 3.2e^{-3}$ | $-5.5, 8.4e^{-5}$ | $-5.7, 5.4e^{-5}$ | $-6.1, 2.6e^{-5}$ |
| Logit-Adjust | $1.9, 0.08$ | $-4.6, 4.4e^{-4}$ | $-5.9, 3.9e^{-5}$ | $-3.5, 3.6e^{-3}$ | $-3.4, 4.7e^{-3}$ | $-4.1, 1.0e^{-3}$ |
| Logit-Adj (post-hoc) | $0.3, 0.75$ | $-6.1, 2.6e^{-5}$ | $-5.7, 5.6e^{-5}$ | $-3.4, 4.3e^{-3}$ | $-3.1, 7.2e^{-3}$ | $-3.4, 4.1e^{-3}$ |

## C.3 COULD DROPS OUTPERFORM FINE-TUNING?

We adopt two kinds of fine-tuning strategies by using the pre-trained model of Cross-Entropy loss in Table 1 and re-train another 2000 iterations with a fixed learning rate (1e-3, 1e-4, 1e-5, 1e-6), including the following two options in Table 5:

**Fine-tuning the whole networks:** named as the row "Cross-Entropy (1e-3)", empirically we observed that the learning rate 1e-3 is consistently better than the others and is reported here.

**Fine-tuning the last layer (DFR):** where learning rate 1e-3 is consistently better than others.

All other settings and model selection criterion remains the same as the vanilla training procedure, except for replacing the training data by the whole validation set. Although these two fine-tuning strategies are beneficial in improving the model performance across each metric and setting over Cross-Entropy loss in Table 1, they still fall largely behind DROPS in most scenarios and require re-training the model on the additional validation set, while DROPS only needs the information of per-class accuracy to decide on the parameters of post-hoc scaling, as also utilized for the model selection of all other baseline methods appeared in Table 1.

Table 5: Performance comparisons on class-imbalanced CIFAR datasets: mean $\pm$ std of averaged class accuracy, $\delta = 1.0$-worst case accuracy, and worst class accuracy of 5 runs are reported: for all three baseline methods, we perform fine-tune of the pre-trained model appeared in Table 1 on the hold out balanced validation set (last 10% of the original balanced training set). Best performed methods in each setting are highlighted. While DROPS only makes use of the validation set to calculate per-class accuracy, without needing to training on samples in the validation set.

| Method | CIFAR-10 (Averaged) | | | CIFAR-100 (Averaged) | | |
|---|---|---|---|---|---|---|
| | $\rho = 10$ | $\rho = 50$ | $\rho = 100$ | $\rho = 10$ | $\rho = 50$ | $\rho = 100$ |
| Cross Entropy (no fine-tune) | 86.51±0.17 | 75.74±0.97 | 69.28±0.94 | 55.59±0.50 | 43.39±0.76 | 37.96±0.27 |
| Cross Entropy (1e-3) | 87.80±0.30 | 82.66±0.24 | 79.68±0.62 | 57.24±0.40 | 48.22±0.40 | 44.80±0.62 |
| DFR$_{\text{Tr}}^{\text{Val}}$ | 87.58±0.25 | 80.32±0.94 | 75.84±0.95 | 57.08±0.52 | 46.68±0.28 | 42.96±0.69 |
| DROPS ($\delta = 0.9$) | 89.17±0.24 | 83.12±0.45 | 80.15±0.50 | 59.69±0.39 | 47.83±0.80 | 43.20±0.78 |
| **Method** | CIFAR-10 ($\delta = 1.0$-worst case acc) | | | CIFAR-100 ($\delta = 1.0$-worst case acc) | | |
| | $\rho = 10$ | $\rho = 50$ | $\rho = 100$ | $\rho = 10$ | $\rho = 50$ | $\rho = 100$ |
| Cross Entropy (no fine-tune) | 79.98±0.13 | 59.48±1.75 | 43.30±3.63 | 27.06±0.53 | 11.24±0.97 | 6.52±0.18 |
| Cross Entropy (1e-3) | 79.14±1.02 | 72.48±0.47 | 69.02±1.31 | 34.38±0.55 | 25.04±0.72 | 21.20±1.24 |
| DFR$_{\text{Tr}}^{\text{Val}}$ | 79.80±0.74 | 70.24±1.52 | 65.10±0.91 | 34.14±0.91 | 22.78±0.52 | 18.86±0.73 |
| DROPS ($\delta = 0.9$) | 86.20±0.34 | 79.40±0.57 | 75.46±0.44 | 44.96±0.52 | 30.12±0.66 | 25.58±0.50 |
| **Method** | CIFAR-10 (Worst) | | | CIFAR-100 (Worst) | | |
| | $\rho = 10$ | $\rho = 50$ | $\rho = 100$ | $\rho = 10$ | $\rho = 50$ | $\rho = 100$ |
| Cross Entropy (no fine-tune) | 78.29±0.86 | 55.32±3.58 | 36.00±6.11 | 9.79±2.57 | 1.38±0.91 | 0.00±0.00 |
| Cross Entropy (1e-3) | 75.48±1.53 | 68.32±1.13 | 65.06±2.04 | 18.60±1.62 | 10.10±1.02 | 7.40±2.58 |
| DFR$_{\text{Tr}}^{\text{Val}}$ | 76.84±1.27 | 66.30±2.24 | 60.96±1.23 | 20.60±1.62 | 11.00±1.41 | 6.80±2.93 |
| DROPS ($\delta = 0.9$) | 82.22±1.30 | 76.33±0.89 | 71.62±1.34 | 25.98±3.95 | 11.65±3.05 | 8.61±2.22 |

