# OpenReview forum: "Distributionally Robust Post-hoc Classifiers under Prior Shifts"
_ICLR.cc/2023/Conference — ICLR 2023 poster_

### Official Review · Reviewer_P4Hw · 2022-10-23

**Confidence:** 3
**Correctness:** 3
**Technical Novelty And Significance:** 3
**Empirical Novelty And Significance:** Not applicable
**Recommendation:** 6

**Clarity, Quality, Novelty And Reproducibility:**

The paper is written very clearly, and uses easy to understand language to demonstrate complicated mathematical ideas. Even though the formulation itself does not seem to be novel, the resulting post-hoc classifier looks novel to me, and is in a very simple form that can facilitate many practical use cases.

**Strength And Weaknesses:**

Strength:

1. The paper is well written. Problem is clearly set up and described. The algorithm is well justified both theoretically and empirically.

2. Starting from a complicated distributional robust formulation, the resulting post-hoc classifier is yet in a very simple form, which would facilitate its use in many practical problems.

3. The proposed method does not require knowledge of the class distribution in the test set, which adds to its applicability compared with a class of methods that uses undersampling to match the distribution between train and test.

Weakness:

1. Even though the authors claimed that their formulation is essentially different from the worst-case optimization (as done in DRO, for example), from what I read, their approach is not very different from DRO. It is still trying to minimize the worst-case loss among a set of distributions centered around some nominal distribution (Eq. (2) and (3)). The conservativeness of the formulation is controlled by some parameter \delta, which is the same with DRO. Could you please clarify what is essentially the difference between your approach and DRO, if any?

2. In Proposition 1, the RHS, should we remove the constraints w.r.t g under max, since we already penalize the constraint violation in the Lagrangian?

3. In Theorem 1, \eta is the ground truth conditional class probabilities, right? Do we assume there is no shift in \eta between train and test? Do we require the estimated \eta to have good properties for your method to work well? I think explanations on the \eta component needs to be added.

4. In the experiments section under label shifts, can you also compare with the DRO solutions that handle class label shifts? Many DRO models reduce to adding a regularizer to the loss function. I am curious about how your method performs compared with those traditional DRO methods.

5. In practice, how should we pick the parameter \delta to ensure an appropriate amount of robustness?

**Summary Of The Paper:**

This paper studied the problem of distributional shifts in classification models and proposed a distributional robust formulation to tackle this problem. The authors showed that the Bayes optimal solution is a post-hoc scaling of the conditional class probabilities, where the scales can be learned via solving a saddle-point problem. They showed the convergence of this post-hoc classifier, and experimentally demonstrated the superior performance of the proposed algorithm on several datasets compared to baselines.

**Summary Of The Review:**

Overall I think this paper is of good quality, and the resulting post-hoc classifier could be potentially valuable to practitioners who are faced with imbalanced classification problem. I like the succinctness of the results regardless of the fact that it was derived from a complicated min-max formulation. The experimental results also look promising. One thing that confuses me is the authors' claim about how their approach differs from DRO (or worst case optimization), which I expect the authors to further clarify. I'd also like to understand in practice, how should we pick the parameter \delta to ensure an appropriate amount of robustness?

---

> ### Author Response · Authors · 2022-11-15
> **Response to Reviewer P4Hw (Part 1)**
>
> **Dear Reviewer P4Hw,**
>
> We sincerely appreciate your detailed comments and positive feedback! We address your concerns as follows.
>
> `1. What is essentially the difference between your approach and DRO, if any?`
>
> **Responses:**
>
> Most of the work on robustness to prior shift considers average [Ref 1] or worst performance [Ref 2]. However, DRO can be generalized to optimize the worst-case expected loss over an uncertainty set of distributions. We consider this general formulation but specifically operate under the problem setting of label or group prior shifts, unlike the earlier work of Duchi and Namkoong (2018). Within this problem of label and group prior shifts, we propose a lightweight post-hoc approach that scales the model prediction only at the evaluation/test time, motivated by the need for finer control over the robustness properties of the model instead of just mean/worst performances. While earlier work in the DRO literature modifies the model training objective itself and uses outer multiplicative reweighting of the instance losses, we work with a pre-trained model and only learn the scalings of the model predictions that are applied at test time. Our paper shows that simple post-hoc adjustments to a model (computed via the $\delta$-robust min-max optimization) can help greatly mitigate the effects of label or group prior shifts.
>
> **References**:
>
> Ref 1: Focus on the Common Good: Group Distributional Robustness Follows. [ICLR 22’]
>
> Ref 2: Distributionally Robust Neural Networks for Group Shifts: On the Importance of Regularization for Worst-Case Generalization. [ICLR 20’]
>
> ` 2. In Proposition 1, the RHS, should we remove the constraints w.r.t g under max, since we already penalize the constraint violation in the Lagrangian?`
>
> **Responses:**
>
> Thanks for noticing this! We changed this constraint to $g\in \Delta_m$.
>
> ` 3. In Theorem 1, eta is the ground truth conditional class probabilities, right? Do we assume there is no shift in eta between train and test? Do we require the estimated eta to have good properties for your method to work well?`
>
> **Responses:**
>
> * (1) Yes, in Theorem 1, $\eta$ is the ground truth conditional-class probabilities P(y|x).
>
> * (2) Yes, the underlying setting assumes that the conditional-class probabilities $P(y|x)$ remain the same between training and evaluation, and that it's only the class/group priors P(y) that changes between train and test.
>
> * (3) In Theorem 3, we offer a upper bound for the performance of the post-shifted classifier, which includes the term $E_x[||\hat{\eta}(x)-\eta(x)||_1]$: a bad estimation of $\eta(x)$ tends to result in a large upper bound due to this term. More specifically, this term measures how well the pre-trained model $\hat{\eta}$ is calibrated, i.e., how well its scores match the underlying conditional-class probabilities. In practice, learning $\eta$ using the training set, and estimating the post-hoc scaling parameters using a held-out validation set works quite well, and yields competitive performance.
>
> ` 4. In the experiments section under label shifts, can you also compare with the DRO solutions that handle class label shifts?`
>
> **Responses**:
>
> Thanks for the suggestion! We included DFR (fine tune the whole network or the last layer re-training) in the label shift setting, where we observe that even though DFR fine-tunes the pre-trained model on the validation set, DROPS still largely outperformed DFR. If the reviewer is interested in comparisons with more DRO methods that could be reduced to adding a regularizer to the loss function, it would be great if you could offer some references and we are willing to include comparisons with them in the paper.
>
> We defer the experiment details and results of DFR performances to the next reply (part 2) due to space limits.

---

> > ### Author Response · Authors · 2022-11-15
> > **Response to Reviewer P4Hw (Part 2)**
> >
> > **Additional reply to your 4th concern:**
> >
> > For the performance of DFR, which is a competitive method in DRO, we tried two kinds of fine-tuning strategies by using the pre-trained model of Cross-Entropy loss in Table 1 for another 2000 iterations with a fixed learning rate, including: fine-tuning the whole network (Table 5: row “Cross-Entropy (1e-3)”) and fine-tuning the last layer (DFR).
> >
> > All other settings and model selection criterion remains the same as the vanilla training procedure, except for replacing the training data with the whole validation set. Although these two variants of DFR are beneficial in improving the model performance across each metric and setting over cross-entropy loss in Table 1, they still fall largely behind DROPS in most scenarios and require re-training the model on the additional validation set, while DROPS only needs the information of per-class accuracy to learn the parameters of post-hoc scaling, as also utilized for the model selection of all other baseline methods appeared in Table 1.
> >
> >
> > Table 5: Performance comparisons on class-imbalanced CIFAR datasets: mean $\pm$ std of Averaged class accuracy, $\delta=1.0$-worst accuracy, and Worst class accuracy of 5 runs are reported: for all three baseline methods, we perform fine-tuning of the pre-trained model appeared in Table1 (in paper) on the hold out balanced validation set (last 10 percentage of the original balanced training set). Best performed methods in each setting are highlighted. While DROPS only makes use of the validation set to calculate per-class accuracy, without needing to train on samples in the validation set.
> >
> > | **Averaged Acc**                   | **CIFAR-10 $\rho=10$** | **CIFAR-10 $\rho = 50$** | **CIFAR-10 $\rho= 100$** | **CIFAR-100 $\rho = 10$** | **CIFAR-100 $\rho = 50$** | **CIFAR-100 $\rho= 100$** |
> > |------------------------------------|------------------------|--------------------------|--------------------------|---------------------------|---------------------------|---------------------------|
> > | **Cross Entropy (no fine-tune)**   | 86.51$\pm$0.17         | 75.74$\pm$0.97           | 69.28$\pm$0.94           | 55.59$\pm$0.50            | 43.39$\pm$0.76            | 37.96$\pm$0.27            |
> > | **Cross Entropy (1e-3)**           | 87.80$\pm$0.30         | 82.66$\pm$0.24           | 79.68$\pm$0.62           | 57.24$\pm$0.40            | **48.22$\pm$0.40**        | **44.80$\pm$0.62**        |
> > | **DFR$_{\text{Tr}}^{\text{Val}}$** | 87.58$\pm$0.25         | 80.32$\pm$0.94           | 75.84$\pm$0.95           | 57.08$\pm$0.52            | 46.68$\pm$0.28            | 42.96$\pm$0.69            |
> > | **DROPS ($\delta=0.9$)**           | **89.17$\pm$0.24**     | **83.12$\pm$0.45**       | **80.15$\pm$0.50**       | **59.69$\pm$0.39**        | 47.83$\pm$0.80            | 43.20$\pm$0.78            |
> > | **$\delta=1.0$-worst Acc**       | **CIFAR-10 $\rho=10$** | **CIFAR-10 $\rho = 50$** | **CIFAR-10 $\rho= 100$** | **CIFAR-100 $\rho = 10$** | **CIFAR-100 $\rho = 50$** | **CIFAR-100 $\rho= 100$** |
> > | **Cross Entropy (no fine-tune)**   | 79.98$\pm$0.13         | 59.48$\pm$1.75           | 43.30$\pm$3.63           | 27.06$\pm$0.53            | 11.24$\pm$0.97            | 6.52$\pm$0.18             |
> > | **Cross Entropy (1e-3)**           | 79.14$\pm$1.02         | 72.48$\pm$0.47           | 69.02$\pm$1.31           | 34.38$\pm$0.55            | 25.04$\pm$0.72            | 21.20$\pm$1.24            |
> > | **DFR$_{\text{Tr}}^{\text{Val}}$** | 79.80$\pm$0.74         | 70.24$\pm$1.52           | 65.10$\pm$0.91           | 34.14$\pm$0.91            | 22.78$\pm$0.52            | 18.86$\pm$0.73            |
> > | **DROPS ($\delta=0.9$)**           | **86.20$\pm$0.34**     | **79.40$\pm$0.57**       | **75.46$\pm$0.44**       | **44.96$\pm$0.52**        | **30.12$\pm$0.66**        | **25.58$\pm$0.50**        |
> > | **Worst Acc**                      | **CIFAR-10 $\rho=10$** | **CIFAR-10 $\rho = 50$** | **CIFAR-10 $\rho= 100$** | **CIFAR-100 $\rho = 10$** | **CIFAR-100 $\rho = 50$** | **CIFAR-100 $\rho= 100$** |
> > | **Cross Entropy (no fine-tune)**   | 78.29$\pm$0.86         | 55.32$\pm$3.58           | 36.00$\pm$6.11           | 9.79$\pm$2.57             | 1.38$\pm$0.91             | 0.00$\pm$0.00             |
> > | **Cross Entropy (1e-3)**           | 75.48$\pm$1.53         | 68.32$\pm$1.13           | 65.06$\pm$2.04           | 18.60$\pm$1.62            | 10.10$\pm$1.02            | 7.40$\pm$2.58             |
> > | **DFR$_{\text{Tr}}^{\text{Val}}$** | 76.84$\pm$1.27         | 66.30$\pm$2.24           | 60.96$\pm$1.23           | 20.60$\pm$1.62            | 11.00$\pm$1.41            | 6.80$\pm$2.93             |
> > | **DROPS ($\delta=0.9$)**           | **82.22$\pm$1.30**     | **76.33$\pm$0.89**       | **71.62$\pm$1.34**       | **25.98$\pm$3.95**        | **11.65$\pm$3.05**        | **8.61$\pm$2.22**         |

---

> > > ### Author Response · Authors · 2022-11-15
> > > **Responses to Reviewer P4Hw (Part 3)**
> > >
> > > ` 5. In practice, how should we pick the parameter $\delta$ to ensure an appropriate amount of robustness?`
> > >
> > > **Responses:**
> > >
> > > We have added an ablation study in Table 3. Suppose the practitioner is interested in $\delta_{\text{eval}}=1.0$ robustness, setting $\delta_{\text{train}}\in [0.5, 1.0]$ frequently reaches best three performances in mean/$\delta=1.0$-worst/worst accuracy by referring to the performance of averaged 5 runs, which indicates that DROPS is not overly sensitive to the parameter $\delta_{\text{train}}$. Setting $\delta_{\text{train}}$ based on the performance on a held-out set at a desired $\delta_{\text{eval}}$ should be able to appropriately improve the model robustness under the prior shift of interest.
> > >
> > > Due to space limits, we report a proportion of the ablation study, and a more detailed version of Table 3 can be found in Table 3 of the revised paper.
> > >
> > > Table 3: Ablation study of DROPS on class-imbalanced CIFAR datasets: mean $\pm$ std of averaged class accuracy, $\delta=1.0$-worst case accuracy, and worst class accuracy of 5 runs are reported. The best three performed $\delta$ for each setting are highlighted.
> > > | **Averaged Acc**           | **CIFAR-100 $\rho = 10$** | **CIFAR-100 $\rho = 50$** | **CIFAR-100 $\rho= 100$** |
> > > |----------------------------|---------------------------|---------------------------|---------------------------|
> > > | **DROPS ($\delta=0.1$)**   | 59.08$\pm$0.38            | **48.24$\pm$0.63**        | 43.37$\pm$0.62            |
> > > | **DROPS ($\delta=0.2$)**   | 59.12$\pm$0.21            | 47.90$\pm$0.52            | **43.58$\pm$0.61**        |
> > > | **DROPS ($\delta=0.3$)**   | 58.62$\pm$0.56            | **48.17$\pm$0.91**        | 42.85$\pm$0.86            |
> > > | **DROPS ($\delta=0.5$)**   | 58.79$\pm$0.21            | 47.84$\pm$0.35            | 42.52$\pm$0.43            |
> > > | **DROPS ($\delta=0.7$)**   | 59.42$\pm$0.30            | 48.04$\pm$0.47            | 43.20$\pm$0.92            |
> > > | **DROPS ($\delta=0.8$)**   | **59.63$\pm$0.10**        | 47.78$\pm$0.96            | **43.44$\pm$0.60**        |
> > > | **DROPS ($\delta=0.9$)**   | **59.69$\pm$0.39**        | 47.83$\pm$0.80            | 43.20$\pm$0.78            |
> > > | **DROPS ($\delta=1.0$)**   | **59.60$\pm$0.63**        | 47.96$\pm$0.86            | 43.23$\pm$1.45            |
> > > | **DROPS ($\delta=1.1$)**   | 59.00$\pm$0.56            | **48.31$\pm$0.46**        | **43.72$\pm$0.55**        |
> > > | **DROPS ($\delta=1.2$)**   | 59.36$\pm$0.37            | 48.02$\pm$1.30            | 43.11$\pm$0.35            |
> > > | **DROPS ($\delta=1.3$)**   | 59.54$\pm$0.75            | 47.86$\pm$1.00            | 42.80$\pm$0.67            |
> > > | **$\delta=1.0$-worst Acc**    | **CIFAR-100 $\rho = 10$** | **CIFAR-100 $\rho = 50$** | **CIFAR-100 $\rho= 100$** |
> > > | **DROPS ($\delta=0.1$)**   | 42.98$\pm$0.39            | 29.62$\pm$0.52            | 24.26$\pm$0.68            |
> > > | **DROPS ($\delta=0.3$)**   | 43.54$\pm$0.43            | 30.18$\pm$1.02            | 25.14$\pm$0.77            |
> > > | **DROPS ($\delta=0.5$)**   | 44.18$\pm$0.34            | 31.10$\pm$0.35            | 25.08$\pm$0.36            |
> > > | **DROPS ($\delta=0.7$)**   | 44.34$\pm$0.90            | 30.30$\pm$0.41            | 25.30$\pm$0.85            |
> > > | **DROPS ($\delta=0.9$)**   | **44.96$\pm$0.52**        | 30.12$\pm$0.66            | **25.58$\pm$0.50**        |
> > > | **DROPS ($\delta=1.0$)**   | 44.86$\pm$1.05            | 31.14$\pm$0.74            | **26.24$\pm$1.88**        |
> > > | **DROPS ($\delta=1.1$)**   | 43.74$\pm$1.33            | **31.18$\pm$0.56**        | **25.84$\pm$0.53**        |
> > > | **DROPS ($\delta=1.2$)**   | **44.92$\pm$0.69**        | **31.28$\pm$1.27**        | 25.06$\pm$0.58            |
> > > | **DROPS ($\delta=1.3$)**   | **45.04$\pm$0.84**        | **31.22$\pm$0.98**        | 25.22$\pm$0.64            |
> > > | **Worst Acc**                | **CIFAR-100 $\rho = 10$** | **CIFAR-100 $\rho = 50$** | **CIFAR-100 $\rho= 100$** |
> > > | **DROPS ($\delta=0.1$)**   | 20.61$\pm$3.24            | 8.25$\pm$2.46             | 6.02$\pm$1.50             |
> > > | **DROPS ($\delta=0.3$)**   | 23.32$\pm$4.39            | 10.90$\pm$3.90            | 7.29$\pm$1.90             |
> > > | **DROPS ($\delta=0.5$)**   | 25.13$\pm$3.05            | 12.52$\pm$2.57            | 6.70$\pm$2.39             |
> > > | **DROPS ($\delta=0.7$)**   | 21.74$\pm$4.97            | 10.89$\pm$2.07            | **9.03$\pm$2.67**         |
> > > | **DROPS ($\delta=0.9$)**   | **25.98$\pm$3.95**        | 11.65$\pm$3.05            | **8.61$\pm$2.22**         |
> > > | **DROPS ($\delta=1.0$)**   | 24.80$\pm$2.85            | 11.86$\pm$2.85            | 7.52$\pm$2.11             |
> > > | **DROPS ($\delta=1.1$)**   | 24.22$\pm$4.83            | **12.81$\pm$2.15**        | **9.42$\pm$1.99**         |
> > > | **DROPS ($\delta=1.2$)**   | **25.77$\pm$4.58**        | **12.89$\pm$2.46**        | 7.27$\pm$2.32             |
> > > | **DROPS ($\delta=1.3$)**   | **26.49$\pm$3.66**        | **12.54$\pm$1.41**        | 8.56$\pm$1.02             |

---

### Official Review · Reviewer_b7xp · 2022-10-25

**Confidence:** 4
**Clarity, Quality, Novelty And Reproducibility:** The paper is clearly written and the …
**Correctness:** 3
**Technical Novelty And Significance:** 3
**Empirical Novelty And Significance:** 3
**Recommendation:** 6

**Strength And Weaknesses:**

Strengths
- Writing is clear
- A simple method that is easy to implement
- Theoretical justification

Weaknesses
- Some experiment details are missing

Detail comments & questions regarding the experiments:

1. What are the sizes of the validation sets? What happens if we fine-tune the model on the validation set?

2. It is mentioned in the tables’ captions that the best performed methods in each setting are highlighted. They are best in terms of what criteria? Did you perform any statistical testing?

3. Figure 1 is concerning. For CIFAR10, why delta_{train}=1 is so good across different eval deltas? If the model is trained based on delta=2, then we should expect it to work best when the evaluation delta is also 2. However, it seems that delta_{train}=1 works best even for evaluation delta = 2. What’s so special about delta_train=1?

4. What’s the performance of DFR for Table 1?

**Summary Of The Paper:**

This work proposes to apply a post-hoc learning process for correcting label (class-prior) shifts between training and test distribution. It utilizes a labelled validation set that has the same label prior distribution as the test data and optimizes a delta-constrained objective (Eq.(1) or (2)). The delta quantifies how much the prior can change from the test/validation class distribution. Using the Lagrangian method, the algorithm alternates between different components. Convergence analysis is provided and experiments on common benchmarks show the superiority of the proposed method, Distributionally RObust PoSt-hoc method (DROPS).

**Summary Of The Review:**

The paper proposes a simple method with theoretical convergence justification. Experiments on common benchmark datasets show that the proposed method can work better than alternative methods. Despite some missing details in the experiments, the results are mostly convincing.

---

> ### Author Response · Authors · 2022-11-15
> **Response to Reviewer b7xp (part 1)**
>
> **Dear Reviewer b7xp,**
>
> We sincerely appreciate your detailed comments and positive feedback! We address your concerns as follows.
>
> `1. What are the sizes of the validation sets? What happens if we fine-tune the model on the validation set? `
>
> **Responses:**
>
> * We adopt a balanced validation set which is the last 10 percent of the original CIFAR training sets. Such a validation set is used for the model selection of all methods. The only information used for *all methods* in Table 1 from the validation set is the per-class accuracy.
>
> * We tried two kinds of fine-tuning strategies by using the pre-trained model of Cross-Entropy loss in Table 1, including:
>
> > * Fine-tuning the whole network (Table 5: row “Cross-Entropy (1e-3)”, re-train another 2000 iterations with a fixed learning rate, (1e-3, 1e-4, 1e-5, 1e-6), 1e-3 is consistently better than the others and is reported here.
>
> > * Fine-tuning last layer (DFR): re-train another 2000 iterations with a fixed learning rate (1e-3, 1e-4, 1e-5, 1e-6), 1e-3 is consistently better than the others and is reported here.
>
> Table 5: Performance comparisons on class-imbalanced CIFAR datasets: mean $\pm$ std of Averaged class accuracy, $\delta=1.0$-worst accuracy, and Worst class accuracy of 5 runs are reported: for all three baseline methods, we perform fine-tune of the pre-trained model appeared in Table1 (in paper) on the hold out balanced validation set (last 10 percentage of the original balanced training set). Best performed methods in each setting are highlighted. While DROPS only makes use of the validation set to calculate per-class accuracy, without needing to train on samples in the validation set.
>
> | **Averaged Acc**                   | **CIFAR-10 $\rho=10$** | **CIFAR-10 $\rho = 50$** | **CIFAR-10 $\rho= 100$** | **CIFAR-100 $\rho = 10$** | **CIFAR-100 $\rho = 50$** | **CIFAR-100 $\rho= 100$** |
> |------------------------------------|------------------------|--------------------------|--------------------------|---------------------------|---------------------------|---------------------------|
> | **Cross Entropy (no fine-tune)**   | 86.51$\pm$0.17         | 75.74$\pm$0.97           | 69.28$\pm$0.94           | 55.59$\pm$0.50            | 43.39$\pm$0.76            | 37.96$\pm$0.27            |
> | **Cross Entropy (1e-3)**           | 87.80$\pm$0.30         | 82.66$\pm$0.24           | 79.68$\pm$0.62           | 57.24$\pm$0.40            | **48.22$\pm$0.40**        | **44.80$\pm$0.62**        |
> | **DFR$_{\text{Tr}}^{\text{Val}}$** | 87.58$\pm$0.25         | 80.32$\pm$0.94           | 75.84$\pm$0.95           | 57.08$\pm$0.52            | 46.68$\pm$0.28            | 42.96$\pm$0.69            |
> | **DROPS ($\delta=0.9$)**           | **89.17$\pm$0.24**     | **83.12$\pm$0.45**       | **80.15$\pm$0.50**       | **59.69$\pm$0.39**        | 47.83$\pm$0.80            | 43.20$\pm$0.78            |
> | **$\delta=1.0$-worst Acc**          | **CIFAR-10 $\rho=10$** | **CIFAR-10 $\rho = 50$** | **CIFAR-10 $\rho= 100$** | **CIFAR-100 $\rho = 10$** | **CIFAR-100 $\rho = 50$** | **CIFAR-100 $\rho= 100$** |
> | **Cross Entropy (no fine-tune)**   | 79.98$\pm$0.13         | 59.48$\pm$1.75           | 43.30$\pm$3.63           | 27.06$\pm$0.53            | 11.24$\pm$0.97            | 6.52$\pm$0.18             |
> | **Cross Entropy (1e-3)**           | 79.14$\pm$1.02         | 72.48$\pm$0.47           | 69.02$\pm$1.31           | 34.38$\pm$0.55            | 25.04$\pm$0.72            | 21.20$\pm$1.24            |
> | **DFR$_{\text{Tr}}^{\text{Val}}$** | 79.80$\pm$0.74         | 70.24$\pm$1.52           | 65.10$\pm$0.91           | 34.14$\pm$0.91            | 22.78$\pm$0.52            | 18.86$\pm$0.73            |
> | **DROPS ($\delta=0.9$)**           | **86.20$\pm$0.34**     | **79.40$\pm$0.57**       | **75.46$\pm$0.44**       | **44.96$\pm$0.52**        | **30.12$\pm$0.66**        | **25.58$\pm$0.50**        |
> | **Worst Acc**                         | **CIFAR-10 $\rho=10$** | **CIFAR-10 $\rho = 50$** | **CIFAR-10 $\rho= 100$** | **CIFAR-100 $\rho = 10$** | **CIFAR-100 $\rho = 50$** | **CIFAR-100 $\rho= 100$** |
> | **Cross Entropy (no fine-tune)**   | 78.29$\pm$0.86         | 55.32$\pm$3.58           | 36.00$\pm$6.11           | 9.79$\pm$2.57             | 1.38$\pm$0.91             | 0.00$\pm$0.00             |
> | **Cross Entropy (1e-3)**           | 75.48$\pm$1.53         | 68.32$\pm$1.13           | 65.06$\pm$2.04           | 18.60$\pm$1.62            | 10.10$\pm$1.02            | 7.40$\pm$2.58             |
> | **DFR$_{\text{Tr}}^{\text{Val}}$** | 76.84$\pm$1.27         | 66.30$\pm$2.24           | 60.96$\pm$1.23           | 20.60$\pm$1.62            | 11.00$\pm$1.41            | 6.80$\pm$2.93             |
> | **DROPS ($\delta=0.9$)**           | **82.22$\pm$1.30**     | **76.33$\pm$0.89**       | **71.62$\pm$1.34**       | **25.98$\pm$3.95**        | **11.65$\pm$3.05**        | **8.61$\pm$2.22**         |

---

> > ### Author Response · Authors · 2022-11-15
> > **Response to Reviewer b7xp (part 2)**
> >
> > **Additional reply for your 1st concern:**
> > All other settings and model selection criterion remains the same as the vanilla training procedure, except for replacing the training data with the whole validation set. Although these two fine-tuning strategies are beneficial in improving the model performance across each metric and setting over cross-entropy loss in Table 1, they still fall largely behind DROPS in most scenarios and require re-training the model on the additional validation set, while DROPS only needs the information of per-class accuracy to learn the parameters of post-hoc scaling, as also utilized for the model selection of all other baseline methods appeared in Table 1.
> >
> > `2. It is mentioned in the tables’ captions that the best-performed methods in each setting are highlighted. They are best in terms of what criteria? Did you perform any statistical testing?`
> >
> > **Responses**:
> >
> > In Table 1, “best” refers to the **best** averaged performance of 5 runs. We further add the statistical testing of Table 1: in Table 4 below (also appeared in the appendix), we included the paired student $t$-test results between each baseline method and DROPS, for each dataset and each metric (mean/$\delta$-worst/worst accuracy), and the inputs of samples of each method for testing are the test accuracies of 5 runs $\times$ 3 imbalance ratio settings). And each cell indicates the **($t$-stastictics and $p$-value)**. It is quite obvious that for most results, there exists negative statistics, meaning that DROPS is significantly (if $p$-value is small enough, i.e., $p<0.05$) better than the given baseline method.
> >
> > Table 4: Paired student $t$-test of the performance comparisons between each baseline method and DROPS: cells in right 6 columns denote (statistics, $p$-value) of the hypothesis testing results between each baseline method and DROPS, the scenario where negative statistics and $p$-value less than 0.05 indicates that DROPS is statistically significant better than the corresponding baseline method.
> >
> > | **Method V.S. DROPS**    | **CIFAR-10 Mean**  | **CIFAR-10 $\delta=1.0$-worst** | **CIFAR-10 Worst**  | **CIFAR-100 Mean** | **CIFAR-100 $\delta=1.0$-worst** | **CIFAR-100 Worst** |
> > |--------------------------|--------------------|-----------------------------|---------------------|--------------------|------------------------------|---------------------|
> > | **Cross Entropy**        | $-11.3, 1.9e^{-8}$ | $-11.1, 2.6e^{-8}$          | $-9.7, 1.4e^{-7}$   | $-7.9, 1.5e^{-6}$  | $-12.9, 3.7e^{-7}$           | $-13.0, 3.3e^{-7}$  |
> > | **Focal**                | $-4.9, 2.2e^{-4}$  | $-15.1, 4.9e^{-10}$         | $-21.2, 4.8e^{-12}$ | $-9.4, 3.6e^{-7}$  | $5.6, 8.8e^{-5}$             | $-4.6, 5.0e^{-4}$   |
> > | **LDAM**                 | $-9.1, 2.8e^{-7}$  | $-9.8, 1.2e^{-7}$           | $-9.2, 2.6e^{-7}$   | $-7.4, 3.1e^{-6}$  | $-6.6, 1.2e^{-5}$            | $-7.4, 3.2e^{-6}$   |
> > | **Balanced-Softmax**     | $-4.1, 1.1e^{-3}$  | $-3.9, 1.6e^{-3}$           | $-3.6, 3.2e^{-3}$   | $-5.5, 8.4e^{-5}$  | $-5.7, 5.4e^{-5}$            | $-6.1, 2.6e^{-5}$   |
> > | **Logit-Adjust**         | $1.9, 0.08$        | $-4.6, 4.4e^{-4}$           | $-5.9, 3.9e^{-5}$   | $-3.5, 3.6e^{-3}$  | $-3.4, 4.7e^{-3}$            | $-4.1, 1.0e^{-3}$   |
> > | **Logit-Adj (post-hoc)** | 0.3, 0.75          | $-6.1, 2.6e^{-5}$           | $-5.7, 5.6e^{-5}$   | $-3.4, 4.3e^{-3}$  | $-3.1, 7.2e^{-3}$            | $-3.4, 4.1e^{-3}$   |
> >
> > `3. Figure 1 is concerning. Why delta_train=1 is so good across different eval deltas? It seems that delta_train=1 works best even for evaluation delta = 2.`
> >
> > **Responses**:
> >
> > This is a great question. We want to first clarify that for the updated results of DROPS in Table 1, we adopted $\eta_g=1/\lambda$ (the step size when performing gradient updates on $g$) for the update of scalar $g$ which yields better performances in most settings, and Figure1 is updated with latest results as well. Regarding your concerns on why the performance of training with $\delta_{\text{train}}=\delta_{\text{eval}}$ may not yield the best on $\delta_\text{eval}$-worst accuracy (to clarify, here $\delta_{\text{eval}}$ refers to the value of $\delta$ used for calculating $\delta$-worst accuracy on the test data), we attribute this to the following aspects:
> >
> > * (3a) In both Figure 1 and Table 1 (paper), for a unified and fair comparison, all the results (reported accuracies) are w.r.t. the model that achieves best $\delta_{\text{eval}}=1$-worst accuracy on the validation set. In other words, even for $\delta_{\text{train}}=2$ curve in Figure 1, the model is selected according to the best $\delta_{\text{eval}}=1$-worst accuracy on the validation set. We do this to study the effect of varying $\delta_{\text{train}}$ on obtaining a model at the desired operating point of $\delta_{\text{eval}}=1$-worst accuracy. Figure 1 also shows how the accuracy will change at various operating points if this model were deployed.

---

> > > ### Author Response · Authors · 2022-11-15
> > > **Response to Reviewer b7xp (part 3)**
> > >
> > > * (3b) For the performance of DROPS under each $\delta_{\text{eval}}$ in Figure 1, the other alternative strategy for model selection by looking at the appropriate $\delta_{\text{eval}}$ performance on the validation set (corresponding to the $\delta_{\text{eval}}$’s on X-axis in Figure 1), could achieve better performance, but it will be unfair when comparing with other baseline methods since we are interested in the robustness of a fixed model under a spectrum of delta-worst measures.
> > >
> > > * (3c) Mismatch between the $\delta_{\text{train}}$ and $\delta_{\text{eval}}$: Note that the post-hoc scalings are learned using a validation set, and as a result, may overfit to the validation set. Therefore, scalings learned with $\delta_{\text{train}}$ are usually not optimal for $\delta_{\text{eval}}=\delta_{\text{train}}$. Thus, we tune $\delta_{\text{train}}$ as part of the model selection procedure using the validation set.
> > >
> > > We have also added the ablation study of DROPS w.r.t. more $\delta_{\text{train}}$ values in Table 3 (in the paper, also given below). It illustrates that DROPS is not overly sensitive to the selection of $\delta_{\text{train}}$, and is generally robust on the full spectrum of $\delta$-worst accuracy. Due to space limits, please refer to the next reply for the detailed results of Table 3!
> > >
> > > `4. What’s the performance of DFR for Table 1?`
> > >
> > > **Responses:**
> > >
> > > This is a great suggestion! Please refer to our reply to your first concern.

---

> > > > ### Author Response · Authors · 2022-11-15
> > > > **Response to Review b7xp (part 4)**
> > > >
> > > > Table 3: Ablation study of DROPS on class-imbalanced CIFAR-100 dataset: mean $\pm$ std of \text{averaged class accuracy}, $\delta=1.0$-worst case accuracy, and worst class accuracy of 5 runs are reported. Best three performed $\delta$ for in each setting are highlighted.
> > > > | **Averaged Acc**           | **CIFAR-100 $\rho = 10$** | **CIFAR-100 $\rho = 50$** | **CIFAR-100 $\rho= 100$** |
> > > > |----------------------------|---------------------------|---------------------------|---------------------------|
> > > > | **DROPS ($\delta=0.1$)**   | 59.08$\pm$0.38            | **48.24$\pm$0.63**        | 43.37$\pm$0.62            |
> > > > | **DROPS ($\delta=0.2$)**   | 59.12$\pm$0.21            | 47.90$\pm$0.52            | **43.58$\pm$0.61**        |
> > > > | **DROPS ($\delta=0.3$)**   | 58.62$\pm$0.56            | **48.17$\pm$0.91**        | 42.85$\pm$0.86            |
> > > > | **DROPS ($\delta=0.4$)**   | 58.98$\pm$0.37            | 47.64$\pm$0.88            | 42.60$\pm$1.15            |
> > > > | **DROPS ($\delta=0.6$)**   | 59.35$\pm$0.27            | 47.44$\pm$0.65            | 42.47$\pm$0.84            |
> > > > | **DROPS ($\delta=0.7$)**   | 59.42$\pm$0.30            | 48.04$\pm$0.47            | 43.20$\pm$0.92            |
> > > > | **DROPS ($\delta=0.8$)**   | **59.63$\pm$0.10**        | 47.78$\pm$0.96            | **43.44$\pm$0.60**        |
> > > > | **DROPS ($\delta=0.9$)**   | **59.69$\pm$0.39**        | 47.83$\pm$0.80            | 43.20$\pm$0.78            |
> > > > | **DROPS ($\delta=1.0$)**   | **59.60$\pm$0.63**        | 47.96$\pm$0.86            | 43.23$\pm$1.45            |
> > > > | **DROPS ($\delta=1.1$)**   | 59.00$\pm$0.56            | **48.31$\pm$0.46**        | **43.72$\pm$0.55**        |
> > > > | **DROPS ($\delta=1.2$)**   | 59.36$\pm$0.37            | 48.02$\pm$1.30            | 43.11$\pm$0.35            |
> > > > | **DROPS ($\delta=1.3$)**   | 59.54$\pm$0.75            | 47.86$\pm$1.00            | 42.80$\pm$0.67            |
> > > > | **$\delta=1.0$-worst Acc** | **CIFAR-100 $\rho = 10$** | **CIFAR-100 $\rho = 50$** | **CIFAR-100 $\rho= 100$** |
> > > > | **DROPS ($\delta=0.1$)**   | 42.98$\pm$0.39            | 29.62$\pm$0.52            | 24.26$\pm$0.68            |
> > > > | **DROPS ($\delta=0.2$)**   | 43.96$\pm$0.18            | 29.98$\pm$1.00            | 25.16$\pm$0.49            |
> > > > | **DROPS ($\delta=0.3$)**   | 43.54$\pm$0.43            | 30.18$\pm$1.02            | 25.14$\pm$0.77            |
> > > > | **DROPS ($\delta=0.4$)**   | 44.16$\pm$0.76            | 30.50$\pm$0.96            | 24.90$\pm$0.88            |
> > > > | **DROPS ($\delta=0.6$)**   | 44.64$\pm$0.35            | 30.12$\pm$0.73            | 24.60$\pm$0.64            |
> > > > | **DROPS ($\delta=0.7$)**   | 44.34$\pm$0.90            | 30.30$\pm$0.41            | 25.30$\pm$0.85            |
> > > > | **DROPS ($\delta=0.8$)**   | 44.16$\pm$0.43            | 30.32$\pm$0.68            | 25.48$\pm$0.55            |
> > > > | **DROPS ($\delta=0.9$)**   | **44.96$\pm$0.52**        | 30.12$\pm$0.66            | **25.58$\pm$0.50**        |
> > > > | **DROPS ($\delta=1.0$)**   | 44.86$\pm$1.05            | 31.14$\pm$0.74            | **26.24$\pm$1.88**        |
> > > > | **DROPS ($\delta=1.1$)**   | 43.74$\pm$1.33            | **31.18$\pm$0.56**        | **25.84$\pm$0.53**        |
> > > > | **DROPS ($\delta=1.2$)**   | **44.92$\pm$0.69**        | **31.28$\pm$1.27**        | 25.06$\pm$0.58            |
> > > > | **DROPS ($\delta=1.3$)**   | **45.04$\pm$0.84**        | **31.22$\pm$0.98**        | 25.22$\pm$0.64            |
> > > > | **Worst Acc**              | **CIFAR-100 $\rho = 10$** | **CIFAR-100 $\rho = 50$** | **CIFAR-100 $\rho= 100$** |
> > > > | **DROPS ($\delta=0.1$)**   | 20.61$\pm$3.24            | 8.25$\pm$2.46             | 6.02$\pm$1.50             |
> > > > | **DROPS ($\delta=0.2$)**   | 22.51$\pm$3.10            | 9.58$\pm$3.34             | 7.40$\pm$1.04             |
> > > > | **DROPS ($\delta=0.3$)**   | 23.32$\pm$4.39            | 10.90$\pm$3.90            | 7.29$\pm$1.90             |
> > > > | **DROPS ($\delta=0.4$)**   | 22.51$\pm$2.87            | 10.79$\pm$2.07            | 8.28$\pm$1.98             |
> > > > | **DROPS ($\delta=0.6$)**   | 24.86$\pm$4.18            | 10.33$\pm$1.92            | 7.43$\pm$2.78             |
> > > > | **DROPS ($\delta=0.7$)**   | 21.74$\pm$4.97            | 10.89$\pm$2.07            | **9.03$\pm$2.67**         |
> > > > | **DROPS ($\delta=0.8$)**   | 24.78$\pm$4.84            | 11.17$\pm$1.54            | 8.21$\pm$2.06             |
> > > > | **DROPS ($\delta=0.9$)**   | **25.98$\pm$3.95**        | 11.65$\pm$3.05            | **8.61$\pm$2.22**         |
> > > > | **DROPS ($\delta=1.0$)**   | 24.80$\pm$2.85            | 11.86$\pm$2.85            | 7.52$\pm$2.11             |
> > > > | **DROPS ($\delta=1.1$)**   | 24.22$\pm$4.83            | **12.81$\pm$2.15**        | **9.42$\pm$1.99**         |
> > > > | **DROPS ($\delta=1.2$)**   | **25.77$\pm$4.58**        | **12.89$\pm$2.46**        | 7.27$\pm$2.32             |
> > > > | **DROPS ($\delta=1.3$)**   | **26.49$\pm$3.66**        | **12.54$\pm$1.41**        | 8.56$\pm$1.02             |

---

### Official Review · Reviewer_xUWa · 2022-10-25

**Confidence:** 3
**Correctness:** 3
**Technical Novelty And Significance:** 2
**Empirical Novelty And Significance:** 2
**Recommendation:** 3

**Clarity, Quality, Novelty And Reproducibility:**

Possible typo in the middle of page 4: To my understanding, the Lagrangean function should eliminate the constraint $D(g, u)\le \delta$, thus, the maximization over $g$ in Proposition 1 should be over $g \in \Delta_m$.

**Strength And Weaknesses:**

I experience certain difficulties in assessing the quality of this paper. I believe that the authors did not clarify well the contributions of the paper.

I am relatively familiar with adversarial training and minimax formulation of equation (3). Theorem 1 of this paper is simply an implication of the Nash equilibrium results, which are abundant in the field of adversarial training and robust learning. For $f$-divergence type set $\mathbb{G}(\delta)$, the results of Duchi and Namkoong (2018) and BenTal et al. (2013) have provided much information about the reweighing schemes and how to solve the problem. The gradient descent-ascent algorithm is also widely used to solve min-max problems. Finally, Theorem 3 provides the guarantee only for the ergodic average $\bar f$, but not for the terminal $f^{(T)}(x)$, thus it is of limited usefulness in practice.

From the perspective of a researcher with moderate experience in adversarial learning, I do not feel that I have learned something new with this paper. I hope that the authors can provide strong justifications for their contributions.

**Summary Of The Paper:**

This paper proposes a classifier that performs well under the shift of class proportions. The approach involves a distributionally robust formulation (a min-max problem) where the weight vector is constrained in a $f$-divergence ball of radius $\delta$. The optimal post-hoc classifier is shown to be a reweighed classifier. The paper contains a gradient descent-ascent type algorithm to find the optimal post-hoc classifier from an initialization.

**Summary Of The Review:**

There is not enough evidence about the novelty and the significance of the results.

---

> ### Author Response · Authors · 2022-11-15
> **Response to Reviewer xUWa**
>
> **Dear Reviewer xUWa,**
>
> We sincerely appreciate your detailed comments! We clarify our main contributions below.
>
> ### Main contributions
>
> While we agree that min-max problems are commonly seen in the robust learning literature, the main novelty in our work comes from:
>
> * The use of the **specific $\delta$-robust formulation** to mitigate prior shifts in long-tail and group-fairness settings;
> * The **light-weight post-hoc procedure** that we propose to solve the optimization problem by making scaling adjustments to a pre-trained model;
> * The **extensive empirical comparisons** demonstrating the efficacy of our proposal in label and group shift applications.
>
> We are **methodologically very different** from the earlier work of Duchi and Namkoong (2018), who consider a robust optimization problem with a generic $f$-divergence constraint on the data generating distribution. Their solution approach uses plain gradient descent to train a model from scratch, with the “maximization over the constraint set” treated as a part of the optimization objective. More specifically, they solve the inner maximization problem to the optimum at every iteration to compute the gradient for model parameters, which isn’t well suited for stochastic optimization (Page 11 in Duchi and Namkoong (2018)). In contrast, we propose a post-hoc procedure that is extremely light-weight and practical, and simply requires making scaling adjustments to a pre-trained model.
>
> Consequently, our theoretical guarantees are also different from Duchi and Namkoong. While they bound the gap between the empirical and population minimizers for the min-max problem, we bound the optimality gap for the scaling adjustments computed by our post-hoc procedure. The reviewer is correct that our proofs build on previous analyses of gradient descent-ascent algorithms. However, a key step in our paper is in adapting those results to provide guarantees for post-hoc scaling of model predictions. We’ll be happy to explore further extending our guarantees to apply to the last iterate instead of the average iterate, and will mention this in our future work.
>
> Finally, we stress that the **prior shift setting is both a fundamental and practically important application** to explore since the label/group shift from training data to test data is quite common in the real-world. Our paper shows that simple post-hoc adjustments to a model (computed via the $\delta$-robust min-max optimization) can help greatly mitigate the effects of such shifts in priors.
>
> We’ll be sure to emphasize our contributions and these differences to prior work in our paper.
>
> **Clarity:**
> Thanks for pointing out the typo! The maximization over $g$ in Proposition 1 should be over $g\in \Delta_m$.

---

### Author Response · Authors · 2022-11-15
**Responses to all reviewers**

**Dear Reviewers and Area Chairs,**

We sincerely appreciate all your time and efforts in reviewing our submitted work! **We have uploaded our revised manuscript with all changes highlighted in red color.** To summarize:

* **Additional Baselines for Class-imbalanced CIFAR**: In Appendix C.3, we include the results of two fine-tuning methods for comparisons: fine-tuning the whole network or the last layer (DFR). Experiment results show that they still fall largely behind DROPS in most scenarios and require re-training the model on the additional validation set, while DROPS only needs the information of per-class accuracy to decide on the parameters of post-hoc scaling, as also utilized for the model selection of all other baseline methods appeared in Table 1.  Thanks **Reviewer b7xp** and **P4Hw** for the suggestion!

* **Improved Empirical Results**: In Table 1 (paper), we updated the results of DROPS since adopting $\eta_g=1/\lambda$ (the step size when performing gradient updates on $g$) for the update of scalar $g$ that yields better performances in most settings in practice. Figure 1 is updated with the latest results as well.

* **How to select $\delta_{\text{train}}$**: In Appendix C.1, we provide an ablation study of DROPS on class-imbalanced CIFAR-100 dataset to illustrate the effect of $\delta_{\text{train}}$. Experiment results demonstrate that DROPS is not overly sensitive to the parameter $\delta_{\text{train}}$. We thank **Reviewer P4Hw** for asking the question!

* **Hypothesis Testing of Performance Comparisons on Class-imbalanced CIFAR**: In Appendix C.2, we include the statistical testing of results in Table 1. Hypothesis testing results demonstrate that DROPS significantly improves over baseline methods under most settings. Thanks Reviewer  **Reviewer b7xp** for the suggestion!

* **Typos**: We revised a typo in Eqn.(4) and Proposition 1 as mentioned by **Reviewer xUWa** and **P4Hw**. Thanks for pointing this out!

Thank you again for your reviews!

Best,

Authors

---

### Decision · Program_Chairs · 2023-01-20

**Decision:**

Accept: poster

**Justification For Why Not Higher Score:**

The paper seems to be neat paper, but I suspect the reviewers didn't find it sufficiently exciting to be on the level of spotlight or oral. The approach seems to be also somewhat related to DRO.

**Justification For Why Not Lower Score:**

In general, this seems to be a clean and neat paper with a simple approach and good empirical performance. The apporach also has a theoretical analysis. Below are some strengths mentioned by the reivewers:

" The paper is well written. Problem is clearly set up and described. The algorithm is well justified both theoretically and empirically.

Starting from a complicated distributional robust formulation, the resulting post-hoc classifier is yet in a very simple form, which would facilitate its use in many practical problems.

The proposed method does not require knowledge of the class distribution in the test set, which adds to its applicability compared with a class of methods that uses undersampling to match the distribution between train and test."

**Metareview: Summary, Strengths And Weaknesses:**


In general, this seems to be a clean and neat paper with a simple approach and good empirical performance. The apporach also has a theoretical analysis. Below are some strengths mentioned by the reivewers:

" The paper is well written. Problem is clearly set up and described. The algorithm is well justified both theoretically and empirically.

Starting from a complicated distributional robust formulation, the resulting post-hoc classifier is yet in a very simple form, which would facilitate its use in many practical problems.

The proposed method does not require knowledge of the class distribution in the test set, which adds to its applicability compared with a class of methods that uses undersampling to match the distribution between train and test."

**Note From Pc:**

if the above contains the word "oral" or "spotlight" please see: "oral" presentation means -> notable-top-5% and "spotlight" means -> notable-top-25%. As stated in our emails, we are disassociating presentation type from AC recommendations

**Summary Of Ac-Reviewer Meeting:**

Two reviewers gave 6, and the third reviewer gave 3. The third reviewer seems to be not able to understand parts of the paper, perhaps due to some mismatch of backgrounds. The AC wrote to the third reviewer but didn't get a reply and thus, there was no video meeting. The AC agrees with the other two reviewers and thinks the paper deserves to be published. Therefore, the AC decided to not take the third reviewer's opinion into account in the decision-making.